# Replication-associated formation and repair of human topoisomerase IIIα cleavage complexes

Liton Kumar Saha [1], Sourav Saha[1], Xi Yang [1], Shar-yin Naomi Huang[1], Yilun Sun [1], Ukhyun Jo[1] & Yves Pommier [1] ✉

Topoisomerase IIIα (TOP3A) belongs to the conserved Type IA family of DNA topoisomerases. Here we report that human TOP3A is associated with DNA replication forks and that a "self-trapping" TOP3A mutant (TOP3A-R364W) generates cellular TOP3A DNA cleavage complexes (TOP3Accs). We show that trapped TOP3Accs that interfere with replication, induce DNA damage and genome instability. To elucidate how TOP3Accs are repaired, we explored the role of Spartan (SPRTN), the metalloprotease associated with DNA replication, which digests proteins forming DNA-protein crosslinks (DPCs). We find that SPRTN-deficient cells show elevated TOP3Accs, whereas overexpression of SPRTN lowers cellular TOP3Accs. SPRTN is deubiquitinated and epistatic with TDP2 in response to TOP3Accs. In addition, we found that MRE11 can excise TOP3Accs, and that cell cycle determines the preference for the SPRTN-TDP2 vs. the ATM-MRE11 pathways, in S vs. G2, respectively. Our study highlights the prevalence of TOP3Accs repair mechanisms to ensure normal DNA replication.

Topoisomerases comprise a family of six enzymes that suppress topological stress and entanglements arising from fundamental nucleic acid transactions including replication, transcription, recombination, chromatin remodeling and DNA repair[1]. Topoisomerases change the three-dimensional structure of DNA by forming covalent bonds as they cut either one strand (type I) or both strands of duplex DNA (type II). Type I topoisomerases are subdivided into enzymes that covalently attach to the 5′- (type IA) or 3′-phosphate (type IB) of the cleaved DNA[1]. All topoisomerases act by a common mechanism of breakage and rejoining of DNA/RNA strand(s). Transient topoisomerase cleavage complexes (TOPccs) are key intermediates for all topoisomerase catalytic reactions. They form by the covalent attachment of catalytic tyrosine residue to the ends of broken DNA strand(s)[2,3]. TOPccs are normally transient and self-reversible. However, their trapping can be highly cytotoxic when they are trapped by antibacterial and anticancer topoisomerase inhibitors as well as endogenous or carcinogenic DNA alterations[1,4].

Human type IA topoisomerases comprise two subtypes, topoisomerase IIIα and IIIβ, designated as TOP3A and TOP3B, respectively[1,5].

TOP3A cleaves one DNA strand generating a TOP3A-linked single strand breaks (termed TOP3Acc) with a TOP3A molecule covalently bound to the 5′-end of the break. After passing another DNA strand through the break, TOP3A reseals the ends of the broken DNA and dissociates from DNA. In human cells, TOP3A functions with the Bloom helicase (BLM) and the scaffolding RecQ-Mediated genome Instability proteins (RMI1 and RMI2), which together form the BTR "dissolvasome" complex that can process HR intermediates to prevent genetic crossovers[6]. The "dissolvasome" has been shown to catalyze the resolution of a broad range of complex substrates, such as D-loops[7], late-replication intermediates[8] and catenated DNA[5]. TOP3A also coordinates with other helicases such as FANCM to suppress sister chromatid exchanges (SCEs) and restart replication[9] and with Polo kinase 1-interacting checkpoint helicase (PICH) for the induction of positive DNA supercoiling[10]. Because of these nuclear activities and additional functions in mitochondria[11], TOP3A is essential in all species examined[12-14]. Additionally, *TOP3A* mutations have been linked to BLM-Syndrome like disorders and diseases in humans[15,16].

---

[1]Developmental Therapeutics Branch & Laboratory of Molecular Pharmacology, Center for Cancer Research, National Cancer Institute, NIH, Bethesda, MD 20892, USA. ✉e-mail: pommier@nih.gov

In *E. coli*, topoisomerase III has been shown recently associated with replication forks and to act on nicked precatenanes formed behind replication forks[17]. In vertebrates, the implication of TOP3A in replication has been proposed[1,18] but not been fully demonstrated. One study suggested that TOP3A, in coordination with other members of BTR complex, contributes to the restart of replication forks by sensing RPA-coated ssDNA[19].

Failure in the reversal of transient TOPccs and their associated DNA breaks results in persistent DNA-protein crosslinks (TOP-DPCs), which, if left unrepaired interfere with DNA metabolism and lead to cell death. Multiple repair pathways are implicated in the repair of TOP1-, TOP2- and TOP3B-DPCs[1,20,21], but until the present report the repair of TOP3Accs had not been revealed. For the other TOP-DPCs, proteolytic processing or debulking of the protein component of the TOP-DPCs can be carried out by SPRTN as well as the proteasome[22,23]. While the proteasome degrades TOP1-, TOP2- and TOP3B-DPCs independently of replication[22,24], SPRTN is a prominent replication-associated protease for TOP1- and TOP2-DPCs as well as for other DPCs[23,25–28]. Depletion of SPRTN causes the accumulation of endogenously trapped TOP1ccs and TOP2ccs in human cells[28], which may explain, in part why SPRTN is essential for embryonic development in mice[25] and why SPRTN downregulation in mouse fibroblasts and hepatocytes increases TOP1ccs. In humans, germline mutations in the *SPRTN* gene cause Ruijs-Aalfs syndrome (RJALS) characterized by premature aging, early onset hepatocellular carcinoma and chromosomal instability[29,30]. In addition to SPRTN, other proteases such as ACRC/GCNA, DDI1 and FAM111A have been shown to degrade the proteinaceous component of DPCs including TOP1- and TOP2-DPCs[31–33].

In humans, two tyrosyl-DNA phosphodiesterase enzymes, TDP1 and TDP2 directly hydrolyze the covalent bonds between the catalytic tyrosines of topoisomerases and the DNA phosphodiester backbone. TDP1 removes 3′-blocking adducts including TOP1-DPCs[34] while TDP2 removes 5′-topoisomerase adducts formed by TOP2- and TOP3B-DPCs[24,35]. The efficient removal of TOP1-, TOP2- and TOP3B-DPCs by TDP1 and TDP2 requires the processing of the topoisomerase protein adducts by the proteasome to expose the tyrosyl-DNA phosphodiester bonds[36,37]. In addition, ZATT (encoded by *ZNF451*), a SUMO E3 ligase, can resolve TOP2-DPCs in coordination with TDP2 without the requirement of proteolytic debulking[38] and several endonucleases have been implicated in the excision of DNA fragments containing TOP-DPCs[1,21]. One such nuclease is MRE11, which has been shown to excise TOP2-DPCs in addition to the proteolytic-TDP pathways[39,40] and been implicated in the processing TOP1-DPCs in-vitro[41].

In this study, we demonstrate that TOP3A is associated with DNA replication in human cells. We elucidate the location and repair of TOP3Accs using a mutated TOP3A (TOP3A-R364W) with diminished resealing activity and show that TOP3Accs are prevalent in replicating DNA during S-phase. We also demonstrate the deleterious consequences of elevated cellular TOP3Accs and the implication of both the SPRTN-TDP2 and the MRE11 pathways in the repair of TOP3A-DPCs.

## Results

### Engineering of a self-trapping TOP3A mutant (R364W) generating TOP3Accs

Previous studies on bacterial E. coli topoisomerase IA (EcTOP1) have demonstrated that mutating an arginine amino acid residue to tryptophane (R321W) proximal to the active site tyrosine residue (Y319), inhibits the DNA resealing step of the enzyme catalytic cycle, resulting in the accumulation of EcTOP1ccs[42]. Likewise, we recently showed that mutating the corresponding residue R338W of human TOP3B results in the accumulation of cellular TOP3Bccs[24]. Sequence alignment revealed that arginine 364 in human TOP3A (corresponding to R321 and R338 in EcTOP1 and human TOP3B, respectively), in the proximity of the catalytic tyrosine residue (Y362), is conserved among type IA topoisomerases (Fig. 1a). Therefore, we set out to mutate R364 in the active

site pocket of human TOP3A (Fig. 1b) to generate a self-trapping mutant of TOP3A to study the functions of TOP3A in human cells and the repair of TOP3Accs. To do this, we generated a plasmid carrying TOP3A with an arginine to tryptophane substitution mutation (R364W) by site-directed mutagenesis.

We reasoned that the R364W mutation should cause the entrapment of TOP3A on the DNA, leading to the accumulation of TOP3Accs. To test this hypothesis, we transiently transfected human HEK293 and HCT116 cells with FLAG-tagged *wild-type* TOP3A (TOP3A-WT) or TOP3A-R364W constructs for 48 h and performed Rapid Approach to DNA Adduct Recovery (RADAR) assays[24,43] to isolate DNA covalently bound protein adducts. After confirming that both plasmids were similarly expressed (-20-fold) relative to mock-transfection (Supplementary Fig. 1a–b), RADAR assays probed with TOP3A antibody revealed that TOP3Accs were readily detectable both in HEK293 and HCT116 cells expressing the TOP3A-R364W plasmid while TOP3A-WT failed to produce detectable TOP3Accs (Fig. 1c).

Immunofluorescence microscopy using pre-extracted U2OS cells ectopically expressing FLAG tagged TOP3A-WT and TOP3A-R364W (Supplementary Fig. 1c) showed multiple chromatin-bound TOP3A foci in cells expressing TOP3A-R364W, with much higher levels than in cells transfected with TOP3A-WT (Fig. 1d–e). We also observed the presence of TOP3A foci in mitochondria (Fig. 1d and Supplementary Fig. 1d), confirming mitochondrial localization of TOP3A[11,16]. These data demonstrate that the self-trapping TOP3A mutant (R364W) binds tightly to chromatin, confirming our results from RADAR assays (Fig. 1c).

### TOP3Accs accumulate prominently in S phase

To determine whether the occurrence of TOP3Accs is cell cycle regulated, we synchronized U2OS cells expressing the self-trapping TOP3A-R364W with double-thymidine block. The cells were released and harvested in G1, S and G2/M phases before performing RADAR assays (Fig. 2a and Supplementary Fig. 2d). Although, a substantial fraction of TOP3A-R364W protein was observed in G1 phase (Supplementary Fig. 2a), TOP3Accs were hardly detectable in G1 (Fig. 2b). Rather, TOP3Accs were detected mostly in S-phase and less in G2/M (Fig. 2b). The levels of TOP3Accs in S-phase cells were 4-fold higher than in G2/M-phase cells.

In addition to the prevalence of TOP3Accs in S-phase, immunofluorescence microscopy revealed the concentration of TOP3A-R364W in active replication foci, as TOP3A signals colocalized with the replication proteins CDC45, RPA1 and RPA2 (Fig. 2c). These results indicate preferential association of TOP3A with replication factories.

This led us to examine whether TOP3A is physically present at DNA replication forks by performing iPOND (isolation of protein on nascent DNA) assays[44]. Like PCNA and CDC45, endogenous TOP3A was present on nascent DNA (Fig. 2d, lane 2) and moved with replisomes as shown by thymidine chase (Fig. 2d, lane 3). Similar results were observed after overexpressing TOP3A-WT and TOP3A-R364W (Fig. 2d, lanes 5-6, and 8-9). In addition, in cells expressing TOP3A-R364W, TOP3A remained detectable after thymidine chase when compared to TOP3A-WT expressing cells, which likely results from trapped TOP3Accs (Fig. 2d–e, compare lanes 6 and 9).

To further characterize the interactions of TOP3A with replication, we asked whether replication inhibition affected TOP3Accs. To this end, we treated the TOP3A-R364W-expressing cells with the inhibitor of replicative DNA polymerases (aphidicolin, APH) for 1 h before harvesting cells and performing RADAR assays (Fig. 2f). APH treatment produced a 70% decrease in the levels of TOP3Accs as compared to drug-free controls (Fig. 2f–g). Similar results were observed when cells were pretreated with the DNA polymerase α inhibitor CD437[45] (Supplementary Fig. 2e). These results demonstrate that TOP3A is released from DNA upon replication inhibition.

To examine whether the accumulation of TOP3Accs in cells transfected with the TOP3A-R364W mutant was related to the known functions of TOP3A in homologous recombination (HR)[5], we downregulated RAD51 by siRNA transfection (Supplementary Fig. 3a). As expected, RAD51 depletion caused an accumulation of cells in G2/M (~2 fold in comparison with siControl) without affecting the S-phase fraction (Supplementary Fig. 3b), consistent with previous observations[46,47]. Yet, we observed no difference in TOP3Accs levels after TOP3A-R364W transfection in RAD51-deficient cells (Supplementary Fig. 3c), indicating that the observed replication defects in cells transfected with TOP3A-R364W is independent of the functions of TOP3A in dissolving HR intermediates.

Altogether, these results demonstrate that TOP3A is coupled with ongoing DNA replication during S-phase.

## Trapped TOP3A (TOP3Accs) hinders DNA replication fork progression

To determine whether and how trapped TOP3Accs affect the progression of DNA replication, we carried out DNA combing assays[48] in U2OS cells ectopically expressing TOP3A-WT and TOP3A-R364W. Nascent DNA was sequentially labeled with CldU and IdU for 30 min each (Fig. 3a). Mutant TOP3A-R364W-expressing cells displayed an overall slower fork speed compared to cells expressing TOP3A-WT. The median fork speeds measured in the TOP3A-R364W and in the

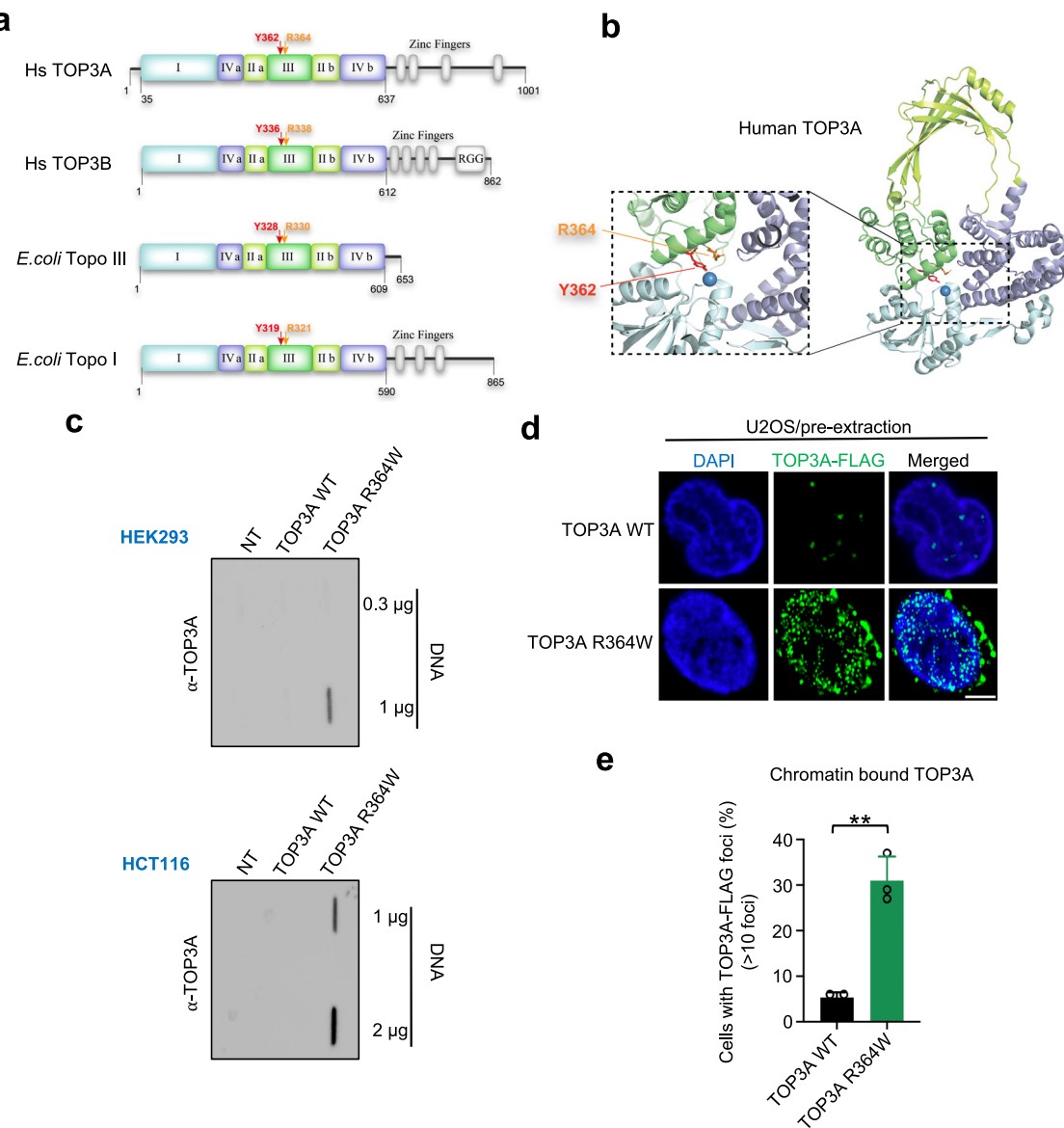

**Fig. 1 | TOP3A forms TOP3Accs in cells transfected with R364W-TOP3A.**
**a** Domain organization alignment of human and bacterial type IA topoisomerases; TOP3A and TOP3B for human, Topo III, and Topo I for *E. coli*. The conserved catalytic tyrosine residues with their amino acid position are indicated by red arrows. Conserved arginine amino acid residues are indicated in orange.
**b** Structure of human TOP3A and ribbon representation of human TOP3A (amino acid [aa] residues 1–637)[10] with the catalytic active site Y362 and the self-trapping mutation site R364. Zoom in on the part of structure where red and orange lines indicate the position of Y362 and R364. **c** Representative slot blot of TOP3Accs detected by RADAR assay in the indicated human cells transfected with the

indicated plasmid constructs for 48 h. TOP3Accs were detected with anti-TOP3A antibody. Indicated amounts of DNA were loaded. **d** U2OS cells expressing FLAG-TOP3A (WT and R364W) were transfected for 48 h and were pre-extracted, fixed and analyzed by confocal microscopy. Representative images are shown. TOP3A foci were detected using anti-FLAG antibody. Scale bars: 10 μm. **e** Quantification of data from experiments as shown in panel D (mean ± SD; at least 100 cells quantified per condition per experiment; n = 3 independent experiments). Quantification of chromatin-bound FLAG-TOP3A foci is shown in cells expressing WT- and R364W-TOP3A. *P*-values were obtained by two-tailed unpaired *t*-test. ***p = 0.0012. Source data are provided as a Source Data file.

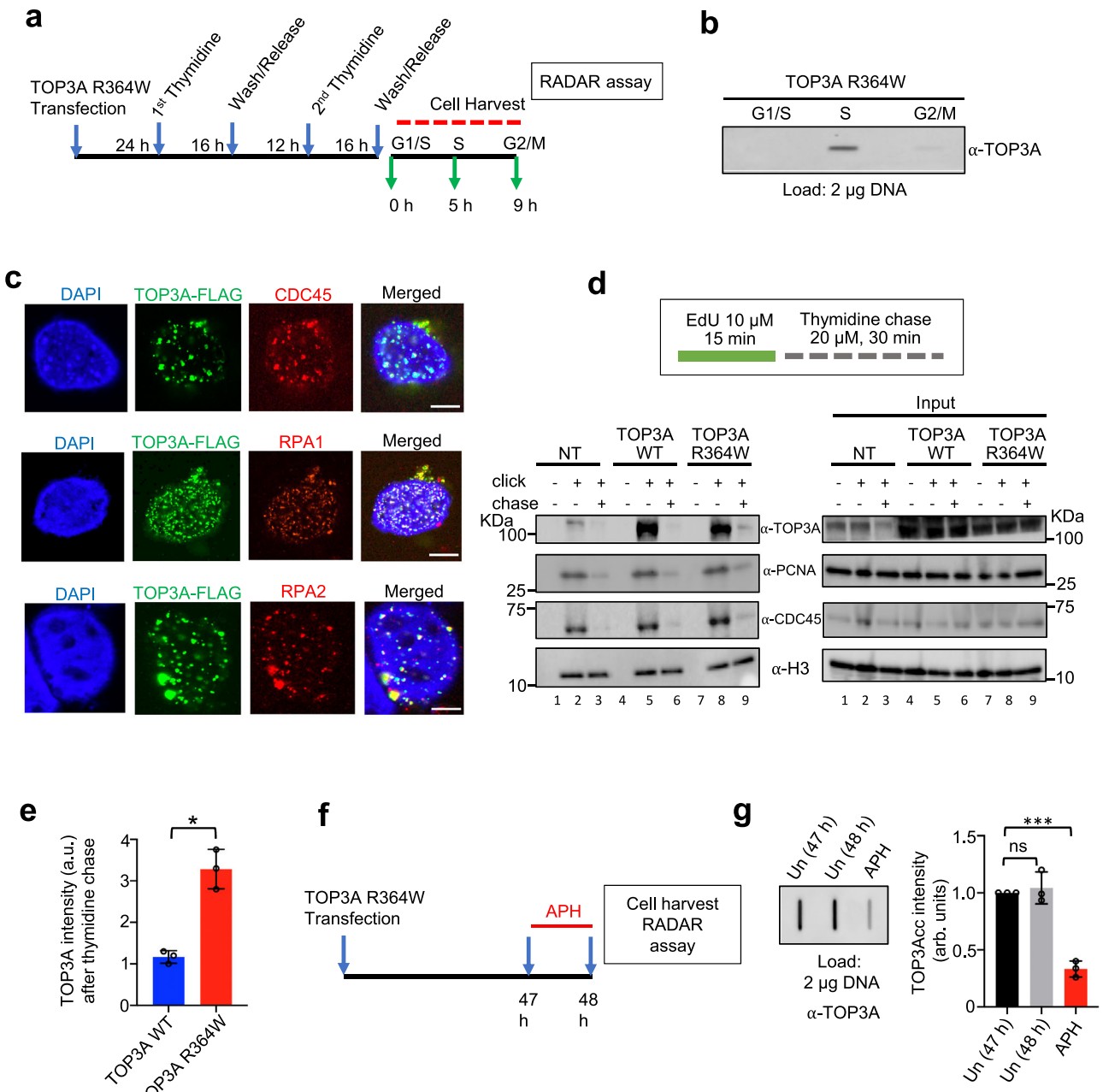

**Fig. 2 | TOP3Accs are predominantly formed in replicative cells. a** Outline of the experimental protocol for transfection, cell synchronization, followed by RADAR assay in U2OS cells with ectopic expression of TOP3A-R364W. Cells at different phases of cell cycle were harvested at the indicated times and protein-DNA adducts were probed by RADAR assay. **b** Representative slot blot for TOP3Accs detection by RADAR assay from cells harvested as indicated in panel A. TOP3Accs were detected with anti-TOP3A antibody. **c** U2OS cells expressing FLAG-tagged TOP3A-R364W after 48 h transfection were pre-extracted, fixed and analyzed by confocal microscopy. Representative images showing TOP3A co-localization with CDC45, RPA1 and RPA2. TOP3A foci were detected using anti-FLAG antibody. Scale bars: 10 μm. **d** Upper panel: workflow of the iPOND experiments; click reactions were performed at the end of the 15 min EdU pulse or following thymidine chase. Lower left panels: lysates form mock-transfected (NT), TOP3A-WT- and TOP3A-R364W-transfected cells were immunoblotted with the indicated antibodies. H3 was used as a loading control. Input samples are shown in the right panels. **e** Quantification of data for the remaining TOP3A signal from 3 independent experiments as shown in Fig. 2D after thymidine chase in cells expressing TOP3A-WT as well as TOP3A-R364W. Chase signal were normalized with respective TOP3A click signal as well as H3 signal. Data represents mean ± SD. *P*-values were obtained by two-tailed unpaired *t*-test with Welch's correction. *$p$ = 0.0105. **f** Protocol for the experiments shown in **g**. TOP3A-R364W-transfected U2OS cells were pretreated with 1 μM aphidicolin (APH) for 1 h before cell harvesting and RADAR assays. **g** Aphidicolin treatment reduces TOP3Accs levels. Left panel: representative slot blot probed with anti-TOP3A antibody. Right panel: quantitation for the three independent experiments as shown on the left panel. Error bar represents mean ± SD. *P*-values were obtained by Ordinary one-way ANOVA with Dunnett multiple comparisons test. ***Adjusted *p*-value = 0.0002, ns not significant (Adjusted *p*-value = 0.8032). Source data are provided as a Source Data file.

TOP3A-WT expressing cells were 1.3 and 1.8 kb/min, respectively (Fig. 3b), indicative of a significant reduction of replication fork velocities in TOP3A-R364W expressing cells.

To assess the stalling or collapse of replication forks in TOP3A-R364W-expressing cells[48], we quantified the number of asymmetrical replication forks in TOP3A-WT and TOP3A-R364W cells (Fig. 3c). Canonical replicons are reflected by symmetrical signals because their divergent forks stained in green (IdU) and red (CldU) move with the same velocity (Fig. 3c)[48]. However, if one of the two forks is stalled prior to or during the pulse labelling, a unidirectional or asymmetrical

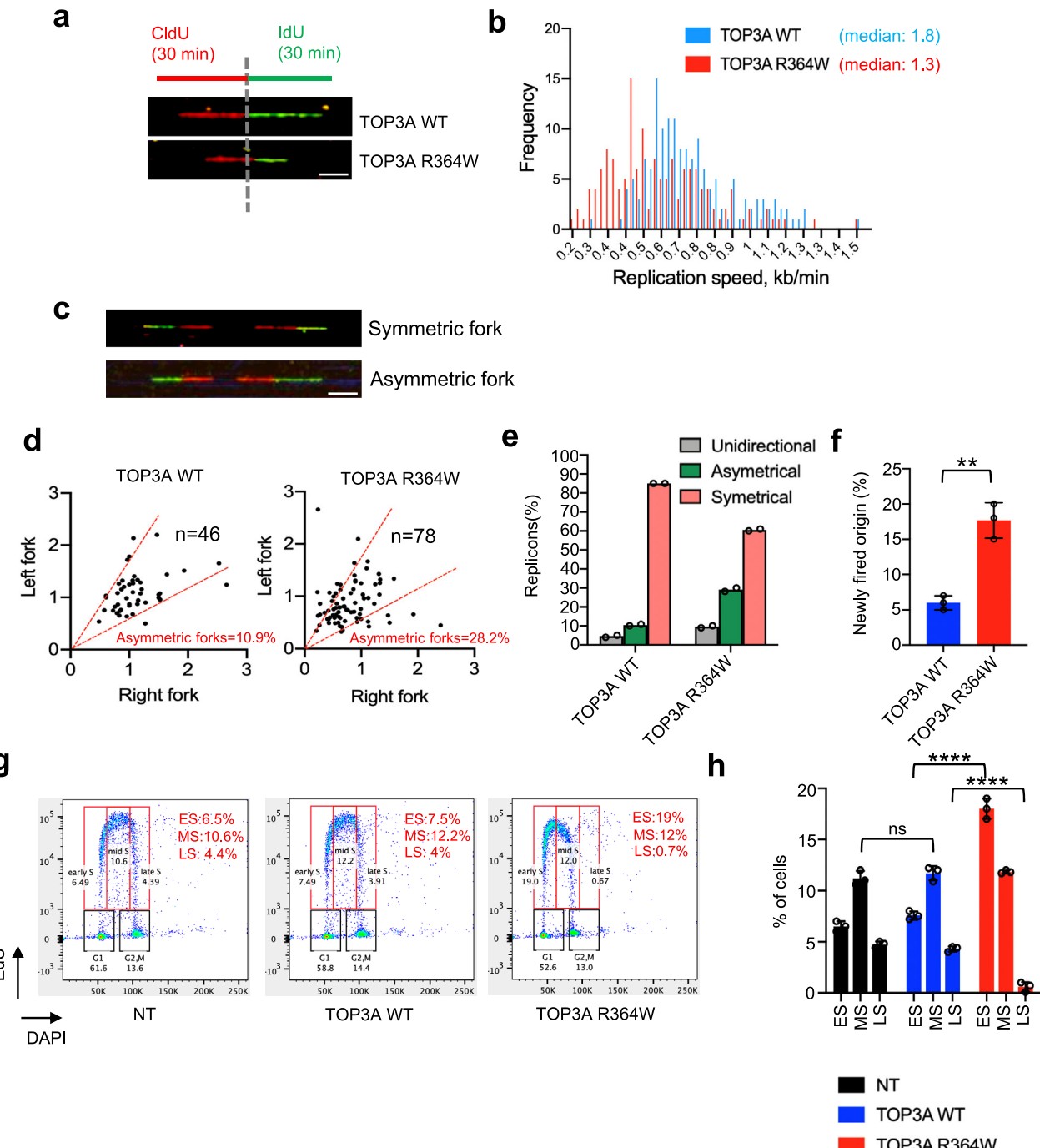

**Fig. 3 | Trapped TOP3Accs alter DNA replication. a** Schematic representation of the DNA combing assay protocol and representative single molecule images. Nascent DNA was labeled with CldU (30 min) followed by IdU (30 min) in U2OS cells transfected with TOP3A-WT and TOP3A- R338W. Scale bars: 100 μm. **b** Histogram showing reduced replication fork speed in U2OS cells expressing R364W-TOP3A. **c** Representative images of a symmetric and an asymmetric replicon observed in DNA combing. Scale bars: 100 μm. **d** Self-trapping TOP3A-R364W induces asymmetric replication forks in U2OS cells. The values of x and y axis represent replication speed (kb/min) of the right fork and left fork, respectively. The percentages of asymmetric forks from the indicated cells are indicated in red. *n* = number of forks analyzed. Asymmetric forks were classified as those forks when the difference between length of left forks and right forks emerging from the same origin was greater than 30%. **e** Histogram showing the percentages of unidirectional (grey), asymmetrical (green) and symmetrical (pink) forks in the indicated cell lines.

Histograms represent the sum of replicons measured in two independent experiments. **f** Newly fired origins are defined as IdU-single labeled (green) fibers. Percentage was calculated from the number of fibers with single green signals divided by the total number of fibers. Error bars indicate the mean value ± SD (*n* = 3 independent experiments). **p = 0.0078 (two-tailed unpaired *t*-test with Welch's correction). **g** Representative flow cytometry plots. U2OS cells transfected as indicated were pulse-labeled with EdU for the last 30 min before harvesting. NT mock transfected. The percentage of replicating (EdU and DAPI-positive) cells measured by BD FACS analysis software is shown in the red text. ES: early-S, MS: mid-S, LS: late-S. **h** A bar plot showing the percentages of EdU positive cells (ES, MS, and LS) from three independent experiments as shown in **g**. Error bars represents mean value ± SD. ns = not significant (Adjusted *p*-value = 0.5304) ****Adjusted *p*-value < 0.0001 (Two-way ANOVA with Tukey's multiple comparisons test). Source data are provided as a Source Data file.

fluorescent signal will be observed, respectively. An elevated number of asymmetries (28.2%) was observed in the TOP3A-R364W-expressing cells (Fig. 3d), a 2-3-fold increase in comparison with TOP3A-WT-expressing cells. The percentage of total stalled forks (unidirectional plus asymmetrical signals among total signals) was higher in TOP3A-R364W cells (about 45%) than in cells transfected with WT-TOP3A (15%). In particular, the level of stalled forks represented by unidirectional signals in TOP3A-R364W-expressing cells (~10%) was about 3-fold higher than in the WT-TOP3A cells (about 3%) (Fig. 3e), indicating that TOP3Accs induce replication fork stalling.

We also analyzed the fraction of IdU single-colored fibers versus the total measured fibers to estimate new origin firing in TOP3A-R364W expressing cells and found ~3-fold increase in origin firing in the TOP3A-R364W cells compared with TOP3A-WT cells (Fig. 3f). Together, the results of the DNA combing experiments indicate that persistent TOP3Accs arrest replicon forks and induce replication origin activation, consistent with replication stress[48,49].

To evaluate the impact of TOP3Accs on global DNA replication, we analyzed DNA replication dynamics and cell cycle distribution by FACS analysis after pulse-exposure (30 min) of cells with EdU. The fraction of early-S phase cells (EdU positive) was significantly increased in TOP3A-R364W cells compared with the TOP3A-WT and non-transfected cells (Fig. 3g–h). Moreover, the S-phase arch appeared different in the R364W cells with a partial collapse of the late-S portion of the arch, indicative of reduced EdU incorporation in the late-S phase cells (Fig. 3g–h). These results show that trapped TOP3Accs interrupt DNA replication both at the fork elongation and origin firing levels, which is reminiscent of the phenotypes of BLM-deficient cells[48]. We therefore conclude that, in cells expressing the TOP3A-R364W mutants, replication fork progression is impaired.

## TOP3A is required for replication fork progression and associated with single-stranded DNA regions associated with fork stalling

To further establish the function of TOP3A at replication forks, we performed additional DNA combing assays after knocking down TOP3A by siRNA. TOP3A-depleted cells showed reduced replication speed as well as increased origin firing and shorter inter-origin distances (Supplementary Fig. 4), which is consistent with the results obtained in the R364W-TOP3A-transfected cells (see Fig. 3a–f). Together, these observations demonstrate that TOP3A activity is required for maintaining normal replication.

Upon replication fork stalling, classically in cells treated with hydroxyurea (HU), replication fork reversal is thought to be a mechanism for fork stabilization[50,51]. A recent study suggested that extensive fork reversal with formation of single-stranded DNA (ssDNA) resulting from 3'-end resection detected by native BrdU labeling was a protective mechanism against genomic instability[52]. We used this BrdU-HU assay (Supplementary Fig. 5a)[52,53] to determine whether TOP3A is important for promoting ssDNA as a potential indicator of replication fork reversal upon replication stress. While in control cells, HU treatment caused the induction of BrdU foci indicative of ssDNA within nascent DNA strands, in TOP3A-depleted cells (by siRNA transfection), we observed a marked reduction in HU-induced ssDNA focus formation (Supplementary Fig. 5b–c). These results suggest that, upon replication stress, TOP3A enables the formation of ssDNA in newly replicated DNA, possibly indicative of facilitation of DNA single-strand resection at reversed replication forks[52,53].

We next examined whether TOP3A is recruited to stalled/reversed replication forks. Immunofluorescence microscopy revealed that TOP3A was specifically enriched in highly dense ssDNA regions due to stalled replication forks in cells transfected with TOP3A-WT (Supplementary Fig. 5d). Notably, TOP3A-R364W-expressing cells showed very similar pattern of co-localization with ssDNA regions as with TOP3A-WT expressing cells (Supplementary Fig. 5d). These observations

suggest the importance of TOP3A in the processing of stalled replication forks.

## Trapped TOP3Accs cause DNA damage and genome instability

To determine whether replication-associated TOP3Accs impair cell viability, we performed colony formation assays with HEK293 cells transfected with mock (NT), TOP3A-WT and TOP3A-R364W. TOP3A-R364W cells exhibited a significant reduction in colony formation efficiency when compared with non-transfected (NT) and WT-TOP3A-transfected cells (Fig. 4a).

Both D-loops, which have been proposed to be major substrates for TOP3A[5,7], and reversed replication forks are associated with the formation of RAD51 filaments/foci following their resection[53]. RAD51 detection by immunofluorescence microscopy showed a marked increase in RAD51 foci in cells expressing TOP3A-R364W in comparison with TOP3A-WT (Fig. 4b–c). Furthermore, we found partial colocalization of TOP3A with RAD51, consistent with the accumulation of TOP3A in D-loop regions and possibly at reversed replication forks (Fig. 4b–d).

Failure to properly coordinate replication fork progression with the topoisomerase-mediated processes that relieve topological constrains may cause fork collapse and DNA double-strand breaks (DSBs). To test this possibility, we measured γH2AX, a classical marker of DNA damage, by immunofluorescence microscopy. Cells expressing TOP3A-R364W showed elevated γH2AX in comparison with cells expressing TOP3A-WT indicating that TOP3A-R364W induces DNA damage (Fig. 4e–f).

Failure in resolving fork-related topological constrains during S phase may result in DNA damage checkpoint activation. Consistent with this possibility, TOP3A-R364W cells exhibited elevated phospho-CHK1 and phospho-ATR (Fig. 4g), indicating activation of the ATR/CHK1 pathway, which typically responds to increased RPA-coated ssDNA caused by DNA damage and replication fork stalling. Activation of the ATM/CHK2 pathway was also observed in TOP3A-R364W cells, as evidenced by the phosphorylation of ATM and CHK2 (Fig. 4g).

Failure to resolve catenanes during replication can lead to the formation of anaphase DNA bridges. These can be either 'bulky' chromatin bridges that can be stained with 4,6-diamidino-2-phenylindole (DAPI) or ultra-fine anaphase bridges (UFBs) that are nucleosome-free and DAPI-negative[54]. To determine whether TOP3A-R364W-expressing cells displayed an altered frequency of chromosomal abnormalities, we analyzed anaphase DNA bridge formation, a widely studied marker of chromosomal instability. Cells expressing TOP3A-R364W showed a significantly elevation of: (i) chromatin bridging (bridges that are DAPI-positive in anaphase), (ii) UFBs (PICH positive in anaphase) and (iii) lagging chromatin (Fig. 4h–i). These results are consistent with the functional role of TOP3A for the faithful segregation of sister chromatids[5,55].

Collectively, these results demonstrate that trapped TOP3Accs lead to replicative DNA damage and genome instability associated with cell cycle checkpoint activation.

## SPRTN is required for the removal of TOP3Accs

As SPRTN repairs DPCs in a DNA replication-coupled manner[28,56], we reasoned that it might respond to and mediate the repair of TOP3Accs. To evaluate this possibility, we checked the SPRTN status of cells expressing TOP3A-R364W. Cellular SPRTN is present in two forms: unmodified and monoubiquitinated. Monoubiquitinated SPRTN has been shown to be deubiquitinated in response to DPCs to allow its recruitment to the DPC sites in chromatin as well as to facilitate its DPC repair activity[27]. Accordingly, we observed the deubiquitylation of endogenous SPRTN in response to TOP3A-R364Wccs (Fig. 5a, lane 3), which implies the activation of SPRTN by trapped TOP3Accs.

To evaluate the role of SPRTN in TOP3Acc repair, we silenced SPRTN by siRNA (Supplementary Fig. 6a) and measured TOP3Accs by

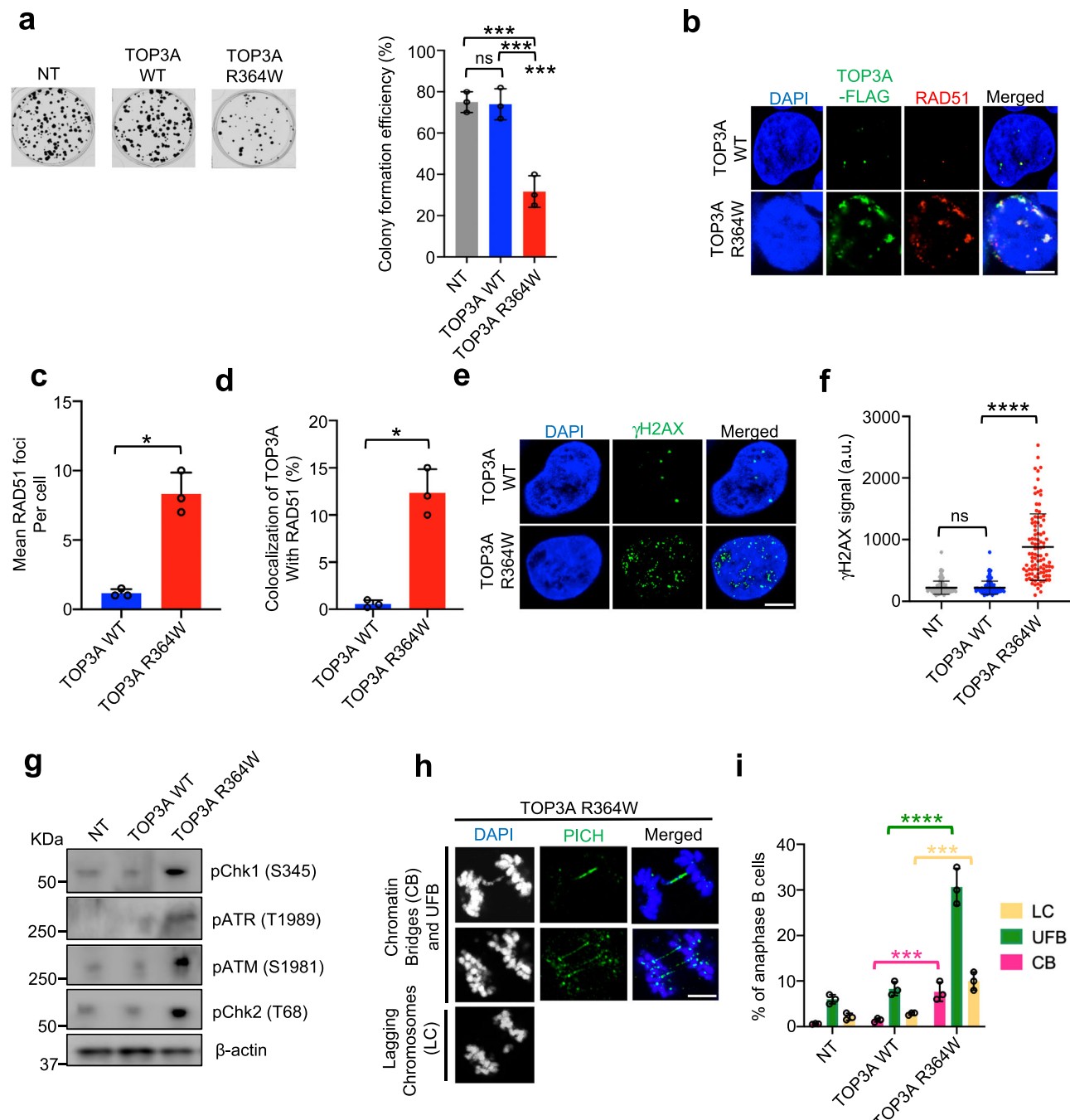

RADAR assay. We observed increased TOP3Accs in SPRTN-deficient cells transfected with TOP3A-R364W (Fig. 5b). Notably, SPRTN-deficient cells also displayed detectable TOP3Accs even after transfection with WT-TOP3A (Fig. 5b). These results demonstrate that SPRTN has an important role in the removal of TOP3Accs. They extend to TOP3A the previously established SPRTN-mediated repair of TOP1ccs and TOP2ccs[25,28,57].

This conclusion was confirmed by showing that ectopic expression of WT SPRTN (Supplementary Fig. 6b) reduced TOP3Acc levels (Fig. 5d). We also found that ectopic expression of the catalytic-dead SPRTN (SPRTN-E112A) and UBZ regulatory domain-deficient SPRTN (SPRTN-ΔUBZ) (Fig. 5c and Supplementary Fig. 6b) compromised the reduction of TOP3Acc levels (Fig. 5d). These data indicate that both the catalytic and regulatory domains of SPRTN determine SPRTN's function in the removal of TOP3Accs from chromatin.

Previous studies suggested the redundant roles of the proteasome and SPRTN in replication-coupled DPC repair[56]. To determine whether the proteasome also plays a role in the processing of TOP3Accs, we treated HCT116 cells transfected with TOP3A-R364W with the proteasome inhibitor, bortezomib. As shown in Supplementary Fig. 7a, inhibiting the proteasome did not impact on TOP3Accs levels, indicating that proteasomal degradation is not required in the repair of cellular TOP3Accs.

We next asked whether SPRTN interacts with TOP3A. Chromatin immunoprecipitation experiments revealed that ectopically expressed EGFP-tagged SPRTN interacted with FLAG-tagged TOP3A in TOP3A-R364W expressing cells (Fig. 5e). The fact that the catalytic-dead SPRTN E112A and regulatory domain-deficient (SPRTN-ΔUBZ) failed to interact with TOP3A (Fig. 5e) suggests that SPRTN interacts with TOP3A through its catalytic and UBZ regulatory domains. However,

**Fig. 4 | Self-trapping R364W-TOP3A causes DNA damage and genome instability. a** Representative images of colony formation assays in HEK293 cells after transfection with TOP3A-R364W, TOP3A-WT or NT; mock transfected (left). Quantitative representation of colony formation assays as shown in left panel (right). Percent of colony formation efficiency was calculated as the percentage of surviving cells after TOP3A plasmids transfection relative to the number of seeded cells of each cell types. Data are provided as means ± SD ($n = 3$ biological replicates). Ordinary one-way ANOVA with Dunnett multiple comparisons test. ns = not significant (Adjusted $p$-value = 0.9825), ***Adjusted $p$-value = 0.0006 (NT vs TOP3A R364W), ***Adjusted $p$-value = 0.0007 (TOP3A WT vs TOP3A R364W). **b** Representative confocal microscopy images of TOP3A and RAD51 immunostaining of U2OS cells transfected with the indicated constructs. TOP3A was detected with anti-FLAG antibody. Scale bar: 10 μm. **c** Quantification of mean RAD51 foci number per nucleus from three independent experiments as shown in panel B. Data represents mean ± SD. Two-tailed unpaired $t$-test with Welch's correction. *$p = 0.0125$. **d** Quantification of the percentage of cells displaying colocalization of TOP3A and RAD51 from three independent experiments as shown in panel B. Data represents mean ± SD. Two-tailed unpaired $t$-test with Welch's correction. *$p = 0.0132$. **e** Representative confocal microscopy images of gamma-γH2AX immunostaining of U2OS cells transfected with the indicated constructs. Scale bar: 10 μm. **f** Trapped TOP3A induces γH2AX in U2OS cells transfected with the indicated constructs. Fluorescence intensities of γH2AX signal per nucleus were analyzed by Image J. Data are the mean ± SD ($n = 102$ cells for NT and TOP3A WT both, and $n = 101$ cells for TOP3A R364W). Two-tailed unpaired $t$-test with Welch's correction. ****$p < 0.0001$ (TOP3A WT vs TOP3AR364W), ns = not significant ($p > 0.9999$). **g** Trapped TOP3A induces DNA damage response (DDR). U2OS cells were transfected as indicated and analyzed by Western blotting with the indicated antibodies. **h** Representative images of TOP3A-R364W-transfected U2OS cells displaying genome instability markers (CB chromatin bridges, LC lagging chromosomes, and UFB ultra-fine bridges) in anaphase stage. Cells were arrested in prometaphase with nocodazole for 3 h and released for 45 min. CB and LC were stained with DAPI, and UFB with PICH. Scale bar: 10 μm. **i** Quantification of CB, LC and UFB in anaphase B (late anaphase) cells in experiments as shown in **h**. Each data point is the mean of 3 independent experiments ±SD. Two-way ANOVA with Dunnett's multiple comparisons test. ***Adjusted $p$-value = 0.0009 (CB: TOP3A-WT vs TOP3A-R364W), ****Adjusted $p$-value < 0.0001 (UFB: TOP3A-WT vs TOP3A-R364W), ***Adjusted $p$-value = 0.0002 (LC: TOP3A-WT vs TOP3A-R364W). Source data are provided as a Source Data file.

the reduced interaction of SPRTN E112A with TOP3A in chromatin is intriguing because the catalytic residue of proteins is not important for the binding of their substrates. Likely, indirect structural changes of SPRTN due to the E112A mutation might lead to loss of its binding with TOP3A in chromatin.

As SUMOylation followed by ubiquitylation is a common step preceding proteolysis[20,22] and has emerged as a signaling mechanism in replication-coupled DPC repair by SPRTN[58], we assessed whether cellular TOP3Accs are ubiquitinated and/or SUMOylated. After transfection with TOP3A, RADAR assay samples were prepared and digested with benzonase to remove the DNA bound to TOPccs[22]. SDS-PAGE and immunoblotting with anti-ubiquitin antibody showed cellular ubiquitination of TOP3Accs in cells transfected with R364W-TOP3A compared to TOP3A-WT-transfected and mock-transfected cells (Fig. 5f). This result is consistent with the implication of SPRTN's UBZ regulatory domain (Fig. 5c) in the modulation of TOP3Acc levels. However, SUMOylation of TOP3Accs was not observed in cells transfected with R364W-TOP3A (Supplementary Fig. 7b), indicating that, by contrast to TOP1ccs[22] (Supplementary Fig. 7b), TOP3Accs are ubiquitylated but not SUMOylated.

Because TRIM41 has been identified as E3 ligase for TOP3Bcc ubiquitylation[24], we tested whether TRIM41 is also an E3 ligase for TOP3Accs. Knocking down TRIM41 (Supplementary Fig. 7c) failed to increase TOP3Accs in cells transfected with R364W-TOP3A (Supplementary Fig. 7d). DUST assay also showed that TRIM41 depletion caused no reduction of ubiquitylated TOP3Accs (Supplementary Fig. 7e), implying that TOP3Accs are not targeted by TRIM41 for their ubiquitylation. Altogether these results demonstrate that SPRTN, in addition to its established functions in removing TOP1ccs and TOP2ccs, also proteolyzes TOP3A-DPCs, and that SPRTN catalytic and regulatory activity is important for this function.

## TDP2-mediated repair of TOP3Accs

In eukaryotes, irreversible TOP2ccs and TOP3Bccs with 5'-phosphotyrosyl DNA linkage are processed by TDP2 while TOP1ccs with 3'-phosphotyrosyl DNA linkage are excised primarily by TDP1[1,59]. As TOP3Accs are 5'-phosphotyrosyl DPCs, we assessed the role of TDP2 in their processing. RADAR assays performed in isogenic TDP2 KO HCT116 cells[60] transfected with TOP3A-R364W showed elevated levels of TOP3Accs in TDP2KO cells in comparison with isogenic *wild-type* cells (Fig. 6a). These results indicate that TDP2 excises cellular TOP3Accs.

We also knocked-down SPRTN in TDP2 KO HCT116 cells using siRNA (Supplementary Fig. 6a) to investigate their genetic interaction for TOP3Accs repair. While cells with either SPRTN or TDP2 deficiency alone displayed higher level of TOP3Accs in comparison with their *wild-type* counterpart, depletion of SPRTN in TDP2 KO cells did not cause further increase in TOP3Accs (Fig. 6a). We confirmed the role of SPRTN and its epistasis with TDP2 in SPRTN KO TK6 cells[61] (Supplementary Fig. 8a). We next asked whether the increased TOP3Accs in SPRTN- and TDP2-deficient HCT116 cells could affect cellular viability. To test this, we performed clonogenic survival assays and found that depletion of SPRTN and TDP2 alone rendered cells more sensitive to TOP3Accs (Fig. 6b). However, SPRTN and TDP2 double depletion caused no further sensitization of cells to TOP3Accs. These results suggest the epistatic relationship between SPRTN and TDP2 in the repair of TOP3Accs, and the sequential action of SPRTN and TDP2, where proteolytic processing (debulking) of TOP3Accs by SPRTN precedes the excision of the remaining TOP3A DPC peptides by TDP2 (Fig. 6c).

## MRE11-mediated repair of TOP3Accs

In yeast and metazoan cells, MRE11 releases and repairs TOP2ccs[39,40,62]. Apart from TOP2ccs, the MRN (MRE11-RAD50-NBS1) complex also repairs TOP1-DPCs in yeast[41,62]. To determine whether MRE11 is involved in TOP3Accs repair, we measured TOP3Accs in MRE11-knockdown HCT116 cells (Supplementary Fig. 6c). MRE11-deficient cells displayed no cell cycle defect (Supplementary Fig. 6e) but increased TOP3Accs (Fig. 6d). Furthermore, MRE11 depletion in TDP2 KO cells caused further increase in TOP3Acc levels in comparison with MRE11 knock-down and TDP2 knock-out cells alone (Fig. 6d and Supplementary Fig. 6c), indicating an additive effect of MRE11 and TDP2 on TOP3Acc removal.

We further evaluated the capability of singly depleted MRE11 or TDP2 cells and MRE11/TDP2 double-depleted cells to act on TOP3Accs-induced DNA damage response by clonogenic survival assay after R364W-TOP3A transfection. As shown in Fig. 6e, colony formation efficiency was compromised by single depletions of MRE11 and TDP2, while concurrent depletion of both MRE11 and TDP2 further reduced cell survival, corroborating the parallel activities of TDP2 and MRE11 in the repair of TOP3Accs (Fig. 6e).

As CtIP functions in coordination with MRE11 and has been implicated in the removal of 5′-TOP2 adducts[63,64], we examined the role of CtIP in TOP3Acc repair, and found that CtIP depletion (Supplementary Fig. 6d) caused an accumulation of TOP3Accs (Fig. 6g). Also, ATM inhibition in CtIP-depleted cells had no additional effect (Fig. 6g). This result is consistent with the known connection between phosphorylated CtIP and MRE11 in DNA end-resection[65].

ATM has been shown to phosphorylate CtIP to promote end-resection during homology-directed repair (HDR) in S/G2 cells[66]. As expected from our CtIP results (Fig. 6g), pharmacological inhibition of ATM and depletion of ATM by siRNA caused an elevation of

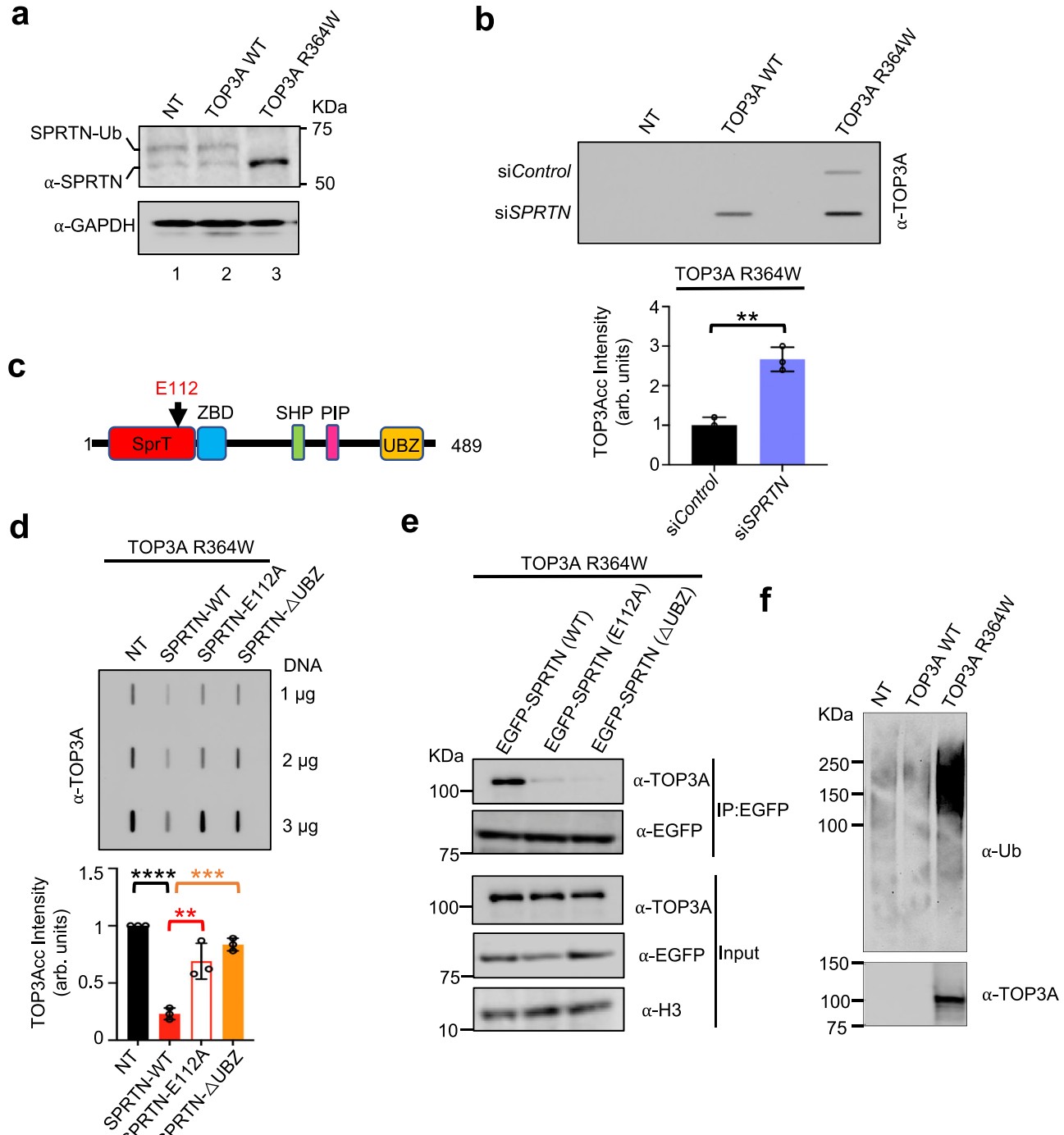

**Fig. 5 | SPRTN promotes the repair of TOP3Accs. a** HCT116 cells were transfected with the indicated constructs. Whole-cell lysates were immunoblotted with the indicated antibodies. GAPDH was used as loading control. NT mock transfected. **b** HCT116 cells were transfected with either si*Control* or si*SPRTN*. After 24 h, they received an additional transfection with the indicated TOP3A constructs and were harvested at 72 h for RADAR assays. Representative slot blot image is shown using anti-TOP3A antibody to detect TOP3Accs. Quantitation is under the image. Error bar represents mean ± SD (*n* = 3 independent experiments). Two-tailed unpaired *t*-test with Welch's correction. **p = 0.0025. **c** Domain architecture of human SPRTN. The catalytic site glutamic acid residue (E112) is noted as black arrow. SprT the metalloprotease domain, ZBD Zinc binding domain, SHP p97 or VCP-binding motif, PIP PCNA interaction peptide, and UBZ ubiquitin-binding zinc finger. **d** Ectopic expression of active full-length SPRTN reduces TOP3Accs. HCT116 cells co-transfected with SPRTN-WT and the indicated mutant plasmids (E112A and ΔUBZ) and TOP3A-R364W were harvested for RADAR assays with anti-TOP3A antibody. A

representative slot blot image is shown. Error bar represents mean ± SD (*n* = 3 independent experiments). Quantitation is plotted under the gel image. Ordinary one-way ANOVA with Dunnett multiple comparison test. ****Adjusted *p*-value ≤ 0.0001 (NT vs SPRTN-WT), **Adjusted *p*-value = 0.0025 (SPRTN-WT vs SPRTN-E112A), ***Adjusted *p*-value = 0.0006 (SPRTN-WT vs SPRTN-ΔUBZ). **e** EGFP immunoprecipitation (IP) of HCT116 cells transfected both with EGFP-SPRTN and FLAG-TOP3A-R364W under denaturing conditions was followed by immunoblotting with FLAG and EGFP antibodies. Cells were lysed and chromatin fractions were immunoprecipitated with S-protein agarose beads. Immunoblotting was performed with the indicated antibodies. **f** Ubiquitination of cellular TOP3Accs. RADAR assay samples were prepared from mock-transfected (NT) U2OS cells or U2OS cells transfected with FLAG-tagged TOP3A-WT or TOP3A-R364W plasmid constructs for 48 h. Equal amounts (3 μg DNA) of RADAR assay samples were digested with benzonase nuclease, ran on SDS-PAGE, and immunoblotted with anti-Ubiquitin (Ub) and anti-TOP3A antibodies. Source data are provided as a Source Data file.

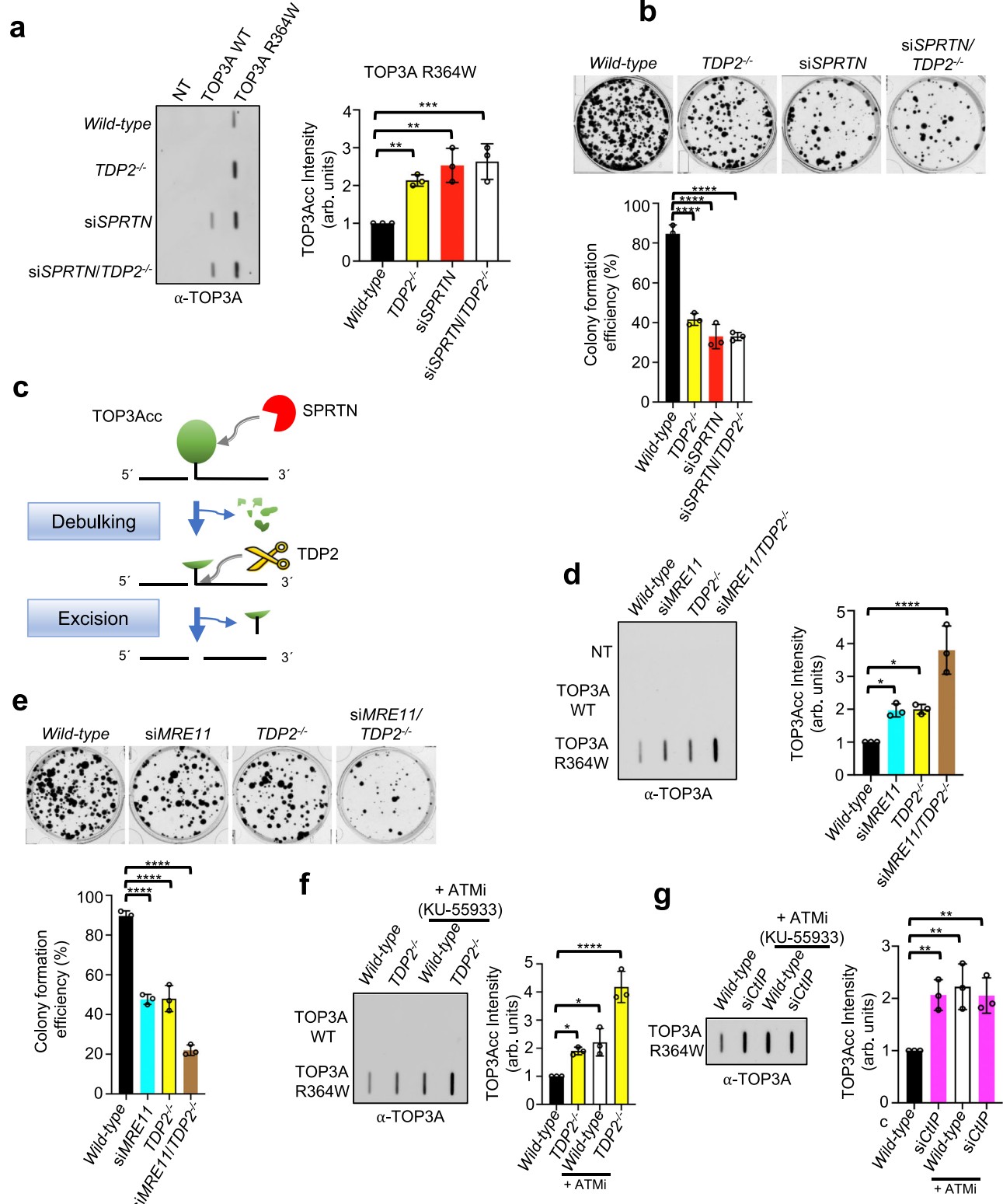

TOP3Accs (Fig. 6f and Supplementary Fig. 8b). Conversely, ATM inhibition in MRE11-depleted cells caused no further increase in TOP3Accs level in comparison with ATM inhibited and MRE11-deficient cells alone (Supplementary Fig. 8c), indicating an epistasis of ATM inhibition and MRE11 depletion for the repair of TOP3Accs. Consistent with the redundant activities of MRE11 and TDP2 in the repair of TOP3Accs, we also found an additive effect of ATM inhibition and TDP2 depletion (Fig. 6f). Altogether these results suggest the existence of parallel pathways for the removal of TOP3Accs. In

one pathway, MRE11, ATM and CtIP act together, and in the other, SPRTN and TDP2 act together.

To explore the coordination and specificity of the MRE11 and TDP2 repair pathways, we examined the effect of MRE11 and SPRTN depletion on TOP3Acc accumulation in S- and G2/M-phase cells in cell cycle synchronized cells. While increased TOP3Accs were mainly observed in S-phase (see Fig. 2), there was a very mild increase in G2/M-phase in SPRTN-depleted cells and the main impact of SPRTN was in S-phase (Fig. 7a, Supplementary Fig. 9a). This result is consistent with

**Fig. 6 | MRE11 and TDP2 repair TOP3Accs in parallel pathways. a** Epistatic relationship between SPRTN and TDP2. Left panel: representative slot-blot. WT and *TDP2*[-/-] HCT116 cells were transfected with the indicated TOP3A plasmid constructs for 48 h and co-transfected with either siControl or siSPRTN for 72 h. TOP3Accs were detected by RADAR assay with anti-TOP3A antibody. Right panel: Quantitation of TOP3Accs from three independent RADAR assays as shown in left panel. NT; mock transfected. Error bar indicates mean ± SD. Ordinary one-way ANOVA with Dunnett multiple comparison test. **Adjusted *p*-value = 0.0083 (Wild-type vs *TDP2*[-/-]), **Adjusted *p*-value = 0.0013 (Wild-type vs si*SPRTN*), ***Adjusted *p*-value = 0.0009 (Wild-type vs si*SPRTN/TDP2*[-/-]). **b** Representative images of colony formation assays in indicated genotypes of HCT116 cells after transfection with TOP3A-R364W. Lower panel: quantification of the data from experiments as shown in upper panel. Clonogenic survival histogram data of each cell types were presented after normalized with their respective colony formation efficiency in TOP3A-WT transfection condition. Error bar represents data mean ± SD (n = 3 independent experiments). Ordinary one-way ANOVA with Dunnett multiple comparison test. ****Adjusted *p*-value = <0.0001 (Wild-type vs *TDP2*[-/-], Wild-type vs si*SPRTN*, Wild-type vs si*SPRTN/TDP2*[-/-]). **c** Model for the coordinated processing of TOP3Accs by SPRTN and TDP2. **d** Additive effects of MRE11 and TDP2 on TOP3Accs. WT and *TDP2*[-/-] HCT116 cells were transfected with the indicated TOP3A plasmid constructs for 48 h and co-transfected with either siControl or siMRE11 for 72 h. Representative slot blot of RADAR assay samples probed with anti-TOP3A antibody. Quantitation is shown to the right of slot-blot images. Error bar indicates the mean value ± SD (n = 3 independent experiments). Ordinary one-way ANOVA with Dunnett multiple comparison test. *Adjusted *p*-value = 0.0380 (Wild-type vs si*MRE11*), *Adjusted *p*-value = 0.0324 (Wild-type vs *TDP2*[-/-]), ****Adjusted *p*-value = <0.0001 (Wild-type vs si*MRE11/TDP2*[-/-]). **e** Representative images of colony formation assays in indicated genotypes of HCT116 cells after transfection with TOP3A-R364W. Lower panel: quantification of the data from experiments as shown in upper panel. Clonogenic survival histogram data of each cell types were presented after normalized with their respective colony formation efficiency in TOP3A-WT transfection condition. Error bar represents data mean ± SD (n = 3 independent experiments). Ordinary one-way ANOVA with Dunnett multiple comparisons test. ****Adjusted *p*-value = <0.0001 (Wild-type vs si*MRE11*, Wild-type vs *TDP2*[-/-], Wild-type vs si*MRE11/TDP2*[-/-]). **f, g** Effects of ATM, CtIP and TDP2 on TOP3Accs. WT and *TDP2*[-/-] HCT116 cells were transfected with the indicated TOP3A plasmid constructs for 48 h. WT cells were co-transfected with either siControl or siCtIP for 72 h. Before harvest, cells were treated with either DMSO or the ATM inhibitor (ATMi) KU-55933 (20 μM) for 2 h. Protein-DNA adducts were isolated by RADAR assay and dot blotted with anti-TOP3A antibody. Representative slot blots are shown with quantitation of TOP3Accs to the right of each panel. Error bar represents data mean ± SD (n = 3 independent experiments). Ordinary one-way ANOVA with Dunnett multiple comparison test. *Adjusted *p*-value = 0.0463 (Wild-type vs *TDP2*[-/-]), *Adjusted *p*-value = 0.0109 (Wild-type vs Wild-type+ATMi), ****Adjusted *p*-value = <0.0001 (*TDP2*[-/-] + ATMi) for (**f**) and **Adjusted *p*-value = 0.0079 (Wild-type vs si*CtIP*), **Adjusted *p*-value = 0.0035 (Wild-type vs Wild-type+ATMi), **Adjusted *p*-value = 0.0084 (Wild-type vs si*CtIP* + ATMi) for **g**. Source data are provided as a Source Data file.

SPRTN expression levels peaking during DNA replication[67]. We also found that the effect of TDP2 on TOP3Accs level was mainly observed in S-phase cells (Fig. 7b, Supplementary Fig. 9a). In contrast, the effect of MRE11 on TOP3Accs was observed mainly in G2/M-phase cells (Fig. 7c, Supplementary Fig. 9a). Together these results lead us to conclude that abortive TOP3Accs are repaired by at least two pathways: SPRTN-TDP2 during S-phase and ATM-CtIP-MRE11 during G2/M (Fig. 7d).

## Discussion

Our study provides new insights on the generation and repair of abortive TOP3A catalytic intermediates (TOP3Accs) in human cells. Using a self-poisoning TOP3A mutant (TOP3A-R364W), we show that TOP3Accs are associated with active replicons and that their stalling leads to DNA damage, genome instability and delayed cellular proliferation. We propose that two repair pathways effectively remove TOP3Accs, SPRTN-TDP2 in S-phase and MRE11-CtIP-ATM in late S- and G2/M-phase (Fig. 7d).

During their catalytic cycles, all topoisomerases act by transiently forming covalent linkages between their active site tyrosines and DNA to reversibly break the DNA backbone, adjust the spatial structure of DNA, and rejoin the DNA break[1]. While TOPccs normally "self-reverse"[1] by completing the topoisomerase catalytic cycle, an important question is what happens when the catalytic cycle of topoisomerases is slowed or impaired. The ways by which cells deal with TOPccs have biological and pharmacological implications, particularly in the context of anticancer and antibacterial drugs and endogenous DNA lesions (abasic sites, mismatches, alkylated bases, DNA breaks), which stabilize or trap TOPccs[1,20].

No small molecule topoisomerase poisons/inhibitors of type IA topoisomerases have been identified thus far to evaluate the cellular consequences of TOP3Accs and uncover the repair pathways for TOP3Accs. Nonetheless, previous studies with the bacterial type IA topoisomerase, E. coli Top1, have demonstrated that individual mutations D111N, D113N, G116S and R321W in the enzyme's TOPRIM domain and active site cause the accumulation of Top1ccs, leading to cell death[42,68,69]. This phenomenon has been attributed to the trapping of E. coli Top1 due to inefficient DNA resealing. A recent study by our group showed that mutating the corresponding arginine residue 338 to tryptophane (R338W) in human TOP3B (corresponding to R321W in E. coli) induces the accumulation of self-trapped TOP3Bccs[24]. Likewise,

our present study shows that mutating the corresponding arginine residue of human TOP3A (R364W) causes self-trapping of TOP3A, thus making it possible to demonstrate the association of TOP3A with DNA replication and study the repair pathways removing TOP3Accs.

A recent study showed the presence of E. coli Top3 in replisomes as evidenced by its colocalization with replication forks using fluorescence microscopy[17]. Likewise, our iPOND analyses reveal the presence of human TOP3A in active replisomes (Fig. 2d), highlighting the importance of TOP3A for normal replication during S-phase in metazoans. Consistently, using immunofluorescence microscopy, we demonstrate the colocalization of TOP3A and replication factory proteins such as CDC45 and RPAs in human cells (Fig. 2c). An interpretation of these results is that, during replication elongation, TOP3A binds to single-strand gaps in the nascent lagging strand bearing Okazaki fragments to remove precatenanes by strand passage in single-stranded DNA segments behind replication forks (Fig. 7d)[1]. This possibility has been proposed recently[70] and is supported by the two recent findings that RPA-coated ssDNA induces the BTR dissolvasome complex to restart stalled replication forks[19] and that E. coli Top3 effectively removes precatenanes[5,17].

Our DNA combing analyses reveal reduced replication fork speed in cells expressing the self-trapping TOP3A-R364W mutant (Fig. 3a–b) and in cells lacking normal levels of TOP3A (Supplementary Fig. 4). This is consistent with the importance of TOP3A for normal replication and with the possibility that accumulation of catenated daughter molecules behind replication forks slows down replication[71]. We also observed increased asymmetric and unidirectional forks in cells expressing the self-trapping TOP3A mutant (Fig. 3d–e), which can be viewed as an increased frequency of stalled replication forks, confirming that TOP3A is necessary to minimize replication fork stalling. The higher frequency of origin firing observed in TOP3A mutant and deficient cells (Fig. 3f and Supplementary Fig. 4) is likely a compensatory mechanism triggered by the slower fork progression to guarantee the full duplication of the genome without increasing the length of the S-phase, as previously observed in Bloom syndrome cells[48]. This may explain why the fraction of EdU-positive cells was not decreased in cells expressing the self-trapping TOP3A-R364W mutant.

In accordance with our iPOND and confocal microscopy findings, the RADAR assays[22,24,43] demonstrated the accumulation of TOP3Accs mainly in S-phase cells (Fig. 2a–b). Remarkably, we observed a marked

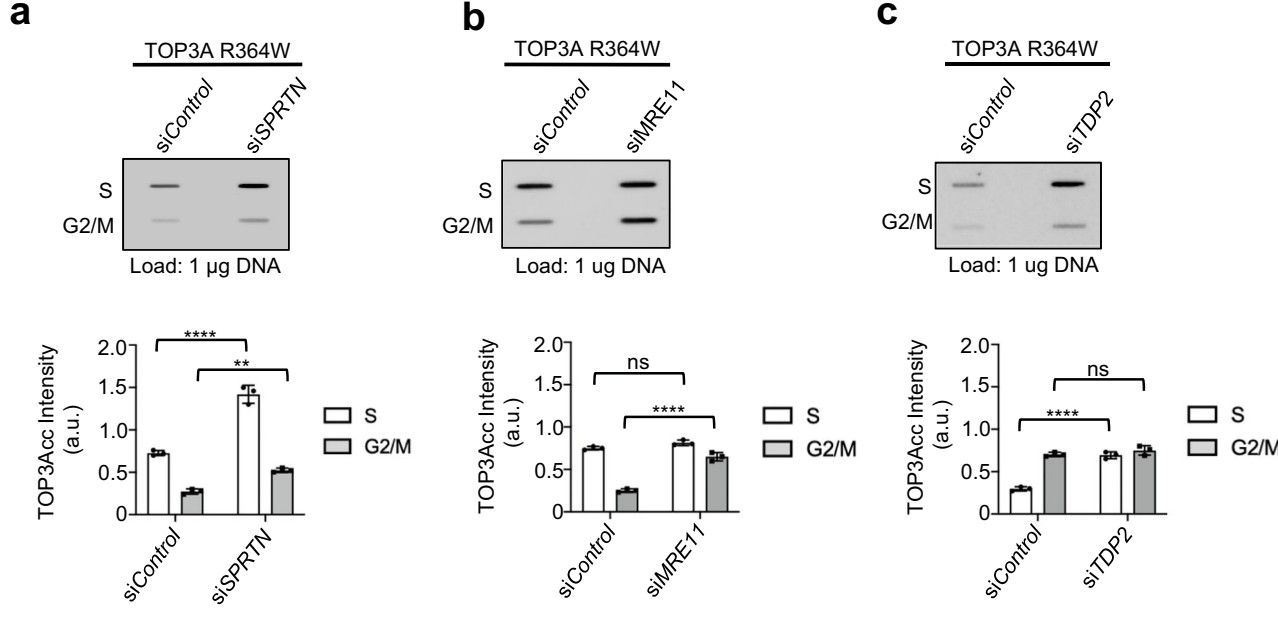

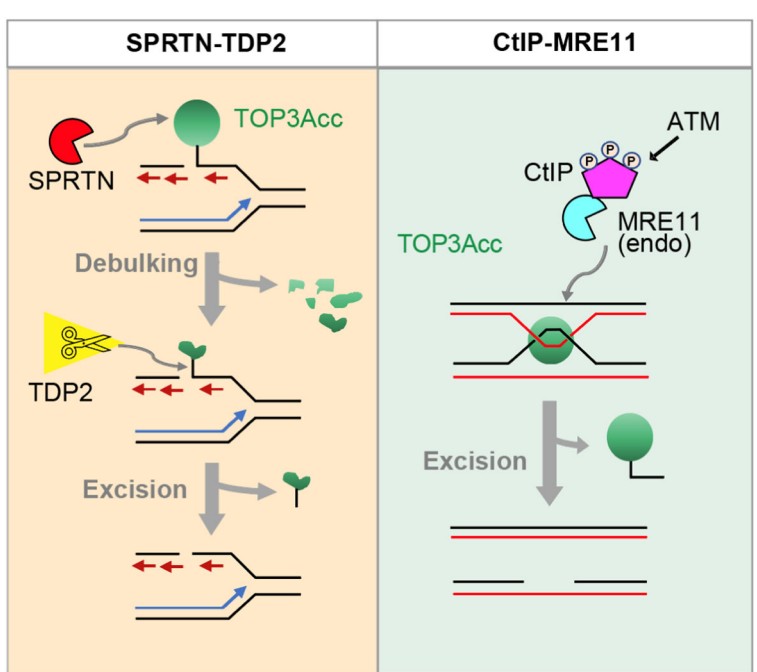

**Fig. 7 | Proposed model for parallel pathways resolving stalled cellular TOP3Accs. a–c** Differential roles of SPRTN, MRE11 and TDP2 for TOP3Accs repair as a function of cell cycle. U2OS cells were transfected with either si*Control* or si*SPRTN* (**a**), si*MRE11* (**b**), si*TDP2* (**c**) followed by co-transfection with TOP3A-R364W. Cells (in S and G2/M phase) were collected after synchronization by double-thymidine block. RADAR assays were performed with anti-TOP3A antibody. Representative slot-blot images are shown. Quantitation from three independent RADAR assays is shown to the bottom of slot blot images. Error bar indicates mean ± SE. Statistical significance was determined by two-way ANOVA with Sidak's multiple comparisons test. ****Adjusted *p*-value ≤ 0.0001(si*Control* vs si*SPRTN*: S-phase, si*Control* vs si*TDP2*: S-phase, si*Control* vs si*MRE11*: G2/M-phase), **Adjusted *p*-value = 0.0016

(si*Control* vs si*SPRTN*: G2/M-phase), [ns]Adjusted *p*-value = 0.0903 (si*Control* vs si*MRE11*: S-phase), [ns]Adjusted *p*-value = 0.3247 (si*Control* vs si*TDP2*: G2/M-phase). Source data are provided as a Source Data file. **d** Left, throughout S-phase (left), trapping of TOP3A at replication forks induces the recruitment of SPRTN. Debulking of TOP3Accs by SPRTN allows TDP2 to excise the TOP3A peptide remnants linked to DNA. Single-ended double-strand breaks (seDSBs) ultimately require homology-directed recombination (HDR) to ensure replication fork progression. Right, at the end of replication and in G2 phase, MRE11 endonucleolytic cleavage removes TOP3Accs from single-strand break sites without needing debulking. MRE11 works together with CtIP. Phosphorylation of CtIP by ATM activates MRE11 for nucleolytic excision of TOP3Accs.

reduction in the levels of TOP3Accs upon acute inhibition of replication by aphidicolin (Fig. 2f–g) and by CD437, a direct inhibitor of DNA polymerase α[45] (Supplementary Fig. 2c). This highlights the implication of active replication as a key factor for the recruitment of TOP3Accs at replication forks (Fig. 7d). We observed that the TOP3Accs accumulation is independent of TOP3A function in dissolving recombination intermediates (Supplementary Fig. 5). In addition to HR, RAD51 plays an important role in protecting replication forks in human cells undergoing replication stress[72]. Whether the fork reversal mechanism by TOP3A is RAD51-dependent, remains to be addressed.

We also found that the accumulation of TOP3Accs strongly affects sister chromatid separation, as demonstrated by our chromosome analyses (Fig. 4h–i). These chromosome segregation defects might arise due to the trapping of our TOP3A mutant leading to faulty/altered TOP3A activity during mitosis[5,10], but could also be a consequence from the replication-associated damage from TOP3A-R364W during S-phase. The defective activity of TOP3A and the accumulation of TOP3Accs could cause sister chromatid entangling while sister chromatids that are still topologically linked during mitosis, which would inevitably lead to chromosome nondisjunction and/or the generation of chromosome breaks followed by DNA damage checkpoint activation (Fig. 4g).

The SPRTN catalytic site (E112) has been implicated in the cleavage of substrates including histones and TOP1 and TOP2 in vitro[28,57]. Moreover, SPRTN catalytic activity has been shown to be important for the repair TOP2ccs in mammalian cells[57]. Our study extends the catalytic activity of SPRTN to the removal of TOP3Accs (Fig. 5c–d). The ubiquitin-binding Zinc finger (UBZ) domain of SPRTN has been proposed to keep SPRTN inactive[27]. Surprisingly, ectopic expression of UBZ-domain-deficient SPRTN failed to reduce the level of TOP3Accs, indicating the potential interaction of post-translational modifications of TOP3Accs and the UBZ domain of SPRTN. Moreover, the in vitro activity of SPRTN has been shown recently to be highest when DPCs are present near ds/ssDNA junction[73]. This is a plausible scenario for TOP3Accs because they can only form in ssDNA segments[5,7,8]. The detection of TOP3Accs within replicons (Fig. 2c) is consistent with the processing of TOP3Accs by SPRTN during S-phase (Fig. 7d, left).

It has been suggested that post-translational modifications (ubiquitination, SUMOylation etc.) of DPCs might facilitate the recruitment of SPRTN[58,74]. Indeed, SUMOylation of TOP1ccs has been proposed as signaling mechanism for TOP1cc and TOP2cc repair by SPRTN[57,74,75]. We also observed the ubiquitination without SUMOylation of TOP3Accs in cells (Fig. 5f and Supplementary Fig. 7b). Additional studies are warranted to determine the ubiquitin ligase for TOP3Accs, and whether other modifications such as PARylation and phosphorylation are associated with the ubiquitylation of TOP3Accs and their recognition by SPRTN.

Our results suggest that SPRTN is insufficient to remove TOP3Accs in the absence of TDP2, as evidenced by an increased level of TOP3Accs in TDP2-depleted cells (Fig. 6a). An epistatic relationship between SPRTN and TDP1 has been suggested for the repair TOP1ccs in mammalian cells[25,28] although WSS1 (yeast SPRTN) and TDP1 appear to act in parallel pathways for TOP1cc repair[76]. Functional relationships between SPRTN and TDP2 remain elusive. In this study, comparable levels of TOP3Accs were observed in si*SPRTN*, *TDP2⁻/⁻* and si*SPRTN/TDP2⁻/⁻* cells after TOP3A-R364W transfection (Fig. 6a), suggesting the coupling of SPRTN and TDP2 for the removal of TOP3A-DPCs (Fig. 7d).

Our study also reveals that cell cycle as a key determinant for repair pathway choice as summarized in Fig. 7. In S-phase where TOP3Accs are prevalent and coincide with SPRTN expression, the debulking-excision SPRTN-TDP2 pathway appears prominent. In G2-M-phase, we propose that CtIP, a phosphorylation substrate of ATM, drives MRE11 activation to remove TOP3Accs by endonucleolytic processing without proteolytic degradation. However, regarding the cell cycle dependency of TOP3Acc repair, we cannot totally exclude that the effect of SPRTN, MRE11 and TDP2 on cell cycle progression may contribute to additional indirect repair defects.

## Methods

### Plasmids and site directed mutagenesis
Human TOP3A-Myc-FLAG cDNA ORF (CAT#: RC208236) clone was purchased from OriGene.

Site-directed mutagenesis was performed using QuikChange II XL site-directed mutagenesis kit (Agilent Technologies) following the manufacturer's protocol, and mutations were confirmed by sequencing. R364W-TOP3A-Myc-FLAG was generated using oligonucleotides: 5′- GGGAAAAATGTTTGTTTCTGTCCAGGGATAGCTGATGTACCCTTG-3′ and 5′- CAAGGGTACATCAGCTATCCCTGGACAGAAACAAACATTTT TCCC-3′. EFGP-SPRTN-E112A were described as previously[61]. EGFP-SPRTN-WT and EGFP-SPRTN-ΔUBZ plasmid was provided by Dr. Lee Zou, Harvard university, USA.

### Cell lines, culture conditions and transfection of expression plasmids
Human HEK293, HCT116 and U2OS cells were cultured in Dulbecco's Modified Eagle Medium (DMEM) (Cat# 084564, Gibco, US) supplemented with fetal bovine serum (10%, Gibco, US), penicillin (100 U/ml), and streptomycin (100 μg/ml, ThermoFischer, US), and maintained at 37 °C under a humidified atmosphere and $CO_2$ (5%). Transient transfection of expression plasmids was carried out using Lipofectamine 3000 reagents (CAT#:L3000015, ThermoFischer, US) according to the manufacturer's protocol for 48 h.

### RADAR assay
TOP3A expression plasmids (FLAG-tagged WT-TOP3A and FLAG-tagged R364W-TOP3A)-transfected cells ($1 \times 10^6$) were washed with PBS and lysed by adding 1 mL DNAzol (ThermoFisher Scientific, CAT#:10503027). Nucleic acids were precipitated following addition of 0.5 mL of 100% ethanol, incubation at −20 °C for 5 min and centrifugation (12,000 x g for 10 min). Precipitates were washed twice in 75% ethanol, resuspended in 200 μL TE buffer, heated at 65 °C for 15 min, followed by shearing with sonication (40% power for 15 s pulse and 30 s rest 5 times). Samples were centrifuged at 21,000 x g for 5 min and the supernatant containing nucleic acids with covalently bound proteins were collected. Nucleic acid containing protein adducts were quantitated, slot-blotted and TOP3Accs were detected with rabbit polyclonal anti-TOP3A antibody (dilution 1:1000, Proteintech, Rosemont, IL, CAT#: 14525-1-AP).

### Isolation of Proteins on Nascent DNA (iPOND)
iPOND was performed as previously described[77]. Briefly, 50 million cells (FLAG-TOP3A-WT, FLAG-TOP3A-R364W and non-transfected cells) labeled with EdU (10 μM) during the last 15 min before collecting cells. Cells were fixed with 2% formaldehyde/PBS for 10 min at room temperature, and the fixation was terminated with 250 mM glycine/PBS. After the fixation, all the procedures were done on ice using pre-chilled buffers. Cells were permeabilized with 0.25% Triton X/PBS for 30 min followed by click reaction (10 mM sodium ascorbate, 2 mM $CuSO_4$, 10 μM biotin-azide in PBS) for 1 h. Cells were resuspended with lysis buffer (1% SDS in 50 mM Tris-HCl pH 8.0) containing protease inhibitor cocktail (2 tablets for 10 mL). Sonication was done by the following settings: pulse 20 s pulse, 40 s pause, amplitude 25%, repeat 10 times (QSONICA Sonicator, ultrasonic processor). After the centrifuge, the supernatant was diluted with PBS by 1:1. Three % volume of each sample was saved as input. The left of the supernatant was incubated with streptavidin-magnet beads for 4 h, and the beads with captured DNA and proteins was washed with lysis buffer followed by 1 M NaCl and another lysis buffer twice. Finally, the beads were incubated with SDS Laemmli sample buffer containing 0.2 M DTT at

98 °C for 10 min. The input and captured proteins were analyzed by western blotting.

## Western blotting

Cells were lysed in 100 µl sodium dodecyl sulfate (SDS) buffer containing Tris–HCl (25 mM, pH 6.5), SDS (1%), β-mercaptoethanol (0.24 mM), bromophenol blue (0.1%) and glycerol (5%). Whole-cell extracts were separated by electrophoresis, transferred onto polyvinylidene difluoride membranes, and blocked in 5% skimmed milk dissolved in Tween-20 (0.1%) containing phosphate buffer saline (PBS). Membranes were incubated with primary antibodies overnight at 4 °C followed by washing in Tween-20 (0.1%) in PBS. Primary antibodies used in this study are as follows: anti-TOP3A (dilution 1:1000, Proteintech, Rosemont, IL, Cat#:14525-1-AP); anti-FLAG (dilution 1:1000, Sigma-Aldrich, Cat# F1804, Clone M2); anti-GAPDH (dilution 1:2000, Cell Signaling Technology, Cat# 2118, Clone 14C10); anti-PCNA (dilution 1:1000, Cell Signaling Technology, Cat# 13110, Clone D3H8P); anti-CDC45 (dilution 1:100, Cell Signaling Technology, Cat# 11881 ); anti-H3 (dilution 1:1000, Cell Signaling Technology, Cat# 9715); anti-EGFP (dilution 1:500, Clontech, Cat# 632380, Clone JL-8); anti-Ub (dilution 1:500, Cell Signaling Technology, Cat# 3936); anti-SPRTN (dilution 1:1000, Atlas Antibodies, Cat# HPA025073); anti-MRE11 (dilution 1:1000, GeneTex, Cat# GTX70212, Clone 12D7); anti-CtIP (dilution 1:1000, Cell Signaling Technology, Cat# 9201, Clone D76F7); anti-β-actin (dilution 1:3000, Sigma-Aldrich, Cat# A5411); anti-TOMM20 (1:500 dilution, Sigma, Cat# HPA011562); anti-SUMO-1 (dilution 1:1000, Cell Signaling Technology, Cat# 4940); anti-SUMO-2/3 (dilution 1:1000, Cell Signaling Technology, Cat# 4971); anti-TRIM41 (dilution 1:1000, Abcam, Cat# ab111580); anti-BLM (dilution 1:1000, Santa Cruz Biotechnology, Cat# sc-365753); anti-RMI1(dilution 1:1000, Thermo Fischer Scientific, Cat# 14630-1-AP); anti-pATR(T1989) (dilution 1:1000, Abcam, Cat# ab223258); anti-pChk1(S345) (dilution 1:1000, Cell Signaling Technology, Cat# 2348, Clone 133D3); anti-pATM(S1981) (dilution 1:1000, Cell Signaling Technology, Cat# 13050, Clone D25E5); anti-pChk2(Thr68) (dilution 1:1000, Cell Signaling Technology, Cat# 2661); anti-TOP1 (dilution 1:1000, BD Biosciences, Cat# 556597); anti-TOP2A (dilution 1:1000, Millipore, Cat# MAB4197); and anti-TOP2B (dilution 1:1000, BD Biosciences, Cat# 611493). Membranes were incubated with anti-mouse IgG ECL, HRP conjugated (dilution 1:4000, GE Healthcare, Cat# NA9310) and anti-rabbit IgG ECL, HRP conjugated (dilution 1:4000, GE Healthcare, Cat# NA9340) at room temperature for 1 h and washed twice and were developed by chemiluminescence with ECL reagent. Images were captured by BioRad ChemiDoc MP Imaging System.

## Immunofluorescence

To visualize FLAG-tagged TOP3A and CDC45, RPA, RPA2, RAD51 and γH2AX foci, cells were plated in 12-well plates on sterilized coverslips one day after transfection. After washing with cold PBS, pre-extraction was performed with CSK buffer (10 mM HEPES-KOH pH7.4, 300 mM sucrose, 100 mM NaCl and 3 mM MgCl$_2$) supplemented with 0.5% Triton X-100 for 10 min on ice followed by fixation in 4% paraformaldehyde in PBS for 20 min at room temperature. Blocking was done in 5% BSA/PBS for 60 min prior to washing with PBS. Coverslips were incubated for overnight with the primary antibodies: anti-FLAG (dilution 1:1000, Sigma-Aldrich, Cat# F1804), Clone M2; anti-CDC45 (dilution 1:100, Cell Signaling Technology, Cat# 11881 S); anti-RPA1 (dilution 1:500, Cell Signaling Technology, Cat# 2267); anti-RPA2 (dilution 1:500, Cell Signaling Technology, Cat# E8X5P); anti-RAD51 (dilution 1:350, Sigma-Aldrich, Cat# PC130); anti-γH2AX(S139) (dilution 1:500, Millipore, Cat# 05-636, Clone JBW301) in 5% BSA/PBS in a humid chamber. After washing with cold PBS, incubation with secondary antibodies: anti-mouse Alexa Fluor 488 (dilution 1:1000, Thermo Fischer Scientific, Cat# A28175) and anti-rabbit Alexa Fluor 568 (dilution 1:1000, Thermo Fischer Scientific, Cat# A-11011) lasted 1 h

in the dark in a humid chamber. Mounting medium with DAPI (VECTASHIELD, Vector Laboratories) was added after the last wash. Images were captured with a Zeiss LSM 880 Airyscan confocal/super resolution microscope with 63x objective lens. Images were analyzed by ImageJ (Fiji, 2020).

## DNA combing

We measured replication fork progression as described with some modifications[61]. Briefly, at 48 h TOP3A plasmids-transfected cells were labeled with 100 µM CldU (Sigma) for 30 min, washed three times with pre-warmed PBS, and labeled with 100 µM IdU (Sigma) for an additional 30 min. After labeling with IdU, cells were immediately washed three times with ice-cold PBS to inhibit DNA replication. Cells were collected, resuspended in PBS, and lysed with lysis buffer (200 mM TrisHCl pH 7.4, 50 mM EDTA, 0.5% SDS). DNA fibers were extracted in agarose plugs and stretched onto silanized coverslips. Combed DNA was dehydrated in an oven at 60 °C for 2 h and denatured with 0.4 M NaOH for 20 min. Samples were washed three times with PBS for 5 min each time on a shaker, dehydrated sequentially in 70, 90, and 100% ethanol for 2 min each and dried at room temperature for 10 min. Samples were blocked with 0.5% BSA in PBS containing 0.1% TritonX-100 (PBST) for 30 min and incubated with rat and mouse anti-BrdU antibodies recognizing CldU and IdU (1 :20 dilution, BD Biosciences, Cat# 347580 and 1: 100 dilution, Abcam, Cat# Ab6326), respectively and a mouse antibody directed against ssDNA (1 :200 dilution, Millipore, Cat# MAB3034) at 4 °C for overnight. After washing with PBST, anti-mouse Cy3 (1:100 dilution, Abcam, Cat# AB97035), anti-rat Cy5 (1:100 dilution, Abcam, Cat# AB6565) and goat anti-mouse BV480 for ssDNA (1:50 dilution, Jackson ImmunoResearch, Cat#115-685-166) were used as secondary antibodies. Slides were scanned with FiberVision Automated Scanner (Genomic Vision). The length of CldU and IdU tracts on single DNA fibers were analyzed using FiberStudio software version 2.0 (Genomic Vision).

## Immunoprecipitation

Cell pellets were incubated on ice for 15 min in pre-extraction buffer (25 mM HEPES, pH 7.4, 50 mM NaCl, 1 mM EDTA, 3 mM MgCl$_2$, 300 mM sucrose, 0.5% Triton-X-100), supplemented with protease and phosphatase inhibitors. After centrifugation (5000 x g, 5 min) and removal of the supernatant, chromatin pellets were resuspended in RIPA buffer (10 mM Tris–HCl pH7.5, 150 mM NaCl, 5 mM EDTA, 0.1% SDS, 1% Triton-X-100, 1% sodium deoxycholate) supplemented with protease and phosphatase inhibitors. Lysates were homogenized and incubated with benzonase for 1 h on a rotator at 4 °C. After centrifugation (15,000 x g, 10 min), the supernatant was collected and used for protein concentration measurement. 300 µg of the extract were resuspended in 200 µl RIPA buffer and 100 µl dilution/washing buffer (10 mM Tris–HCl pH7.5, 150 mM NaCl, 5 mM EDTA) supplemented with protease and phosphatase inhibitors. Samples were then incubated with 2 µg of anti-EGFP antibody (Clonetech, JL-8) on a rotator overnight at 4 °C. Protein G agarose beads were washed three times for 5 min with washing buffer and incubated with the samples for 3 h on a rotator at 4 °C. After three washes, proteins were eluted with loading buffer and incubated for 5 min at 95 °C on a thermomixer. After centrifugation at 15,000 x g (1 min), supernatants were transferred to new tubes, and the proteins separated by electrophoresis.

## siRNA transfections

siRNAs were obtained from Horizon Discovery (Dharmacon) and transfected using Lipofectamine RNAiMAX (Invitrogen) according to manufacturer's protocol. The non-targeting siRNA (siControl) was obtained from Horizon Discovery (Dharmacon) and used as the control. Cells were plated for assays 72 h later. The following siRNA were used: Control siRNA (Control Pool, D-001206-13-5); TOP3A siRNA

(SMARTPool, L-005279-00-0005); RAD51 siRNA (SMARTPool, L-003530-00-0005), SPRTN siRNA (SMARTPool, L-015442-02-0005); MRE11 siRNA (SMARTPool, E-009271-00-0005); CtIP siRNA (SMART-Pool, E-011376-00-0005); and TDP2 siRNA (SMARTPool, E-017578-00-0005).

## Cell cycle analysis

Cell cycle analysis was performed using the Click-iT EdU (5-ethynyl-2′-deoxyuridine) Alexa Fluor 488 Flow Cytometry Assay Kit (Thermo Fisher Scientific) and DAPI Stain (Thermo Fisher Scientific) according to the manufacturer's instructions. Briefly, U2OS cells were seeded onto 6 well plates at a seeding density of $0.3 \times 10^6$ cells per dish. After 48 h of transfection with plasmids, cells were pulsed with Click-iT EdU for 1 h. After harvesting the cells through trypsinization, cells were fixed with Click-iT fixative and permeabilized with a saponin-based permeabilization agent. The Click-iT reaction was then performed for 30 min at room temperature. Cells were then washed with wash buffer and stained for DNA content analysis with DAPI, a DNA-selective dye. Finally, cell cycle analysis was performed using a BD LSRFortessa cell analyzer with FACSDiva software (version 6.2). Data were analyzed using the FlowJo 10.8.1 software.

## Colony formation assay

U2OS cells were transfected with R364W-TOP3A, WT-TOP3A or mock-transfection reagent (NT). 48 h after transfection, cells were harvested and serially diluted with fresh media and plated in 6-well plates (200-300 cells/ well) in triplicate. After 12 days, colonies were fixed, stained with crystal violet, and colonies were counted manually. Data were normalized to the number of seeding cells at Day 0 of each cell genotypes.

## Mitotic analysis

Mitotic abnormalities defects were analyzed as described previously[55]. In brief, for mitotic analysis, transfected cells were seeded onto $22 \times 22$ mm glass coverslips in 6-well plates. Next day, cells were treated with nocodazole for 3 h for the induction of a prometaphase arrest and were subsequently released into fresh media. After 45 min, cells were fixed with 4% paraformaldehyde containing 0.2% Triton X-100 in PBS for 20 min. Fixed cells were incubated with antibody specific to PICH overnight and mounted with Vectashield mounting medium with DAPI. Images were captured by Zeiss LSM 880 Airyscan confocal/super resolution microscope with 63x objective lens and percentage of anaphase cells with chromatin bridges, ultrafine bridges and lagging chromatin were scored.

## Detection of ubiquitinated and SUMOylated TOP3Accs (DUST)

For detection of ubiquitinated and SUMOylated TOP3Accs, we performed DUST assay as described previously[22]. Briefly, nucleic acids and covalent protein-nucleic acid adducts were recovered from FLAG-tagged R364W-TOP3A transfected cells using the RADAR assay. 10 μg of each RADAR assay sample was digested with 250 units benzonase nuclease (EMD Millipore, 100 units/μl) in the presence of 5 mM $CaCl_2$, followed by SDS-PAGE electrophoresis for immunodetection of total TOP3Accs and ubiquitinated and SUMOylated TOP3Accs by probing with rabbit anti-TOP3A (dilution 1:1000, Proteintech, Rosemont, IL, CAT#: 14525-1-AP), mouse anti-ubiquitin antibody (dilution 1:500, Cell Signaling Technology, Cat# 3936), rabbit anti-SUMO-1 (dilution 1:1000, Cell Signaling Technology, Cat# 4940) and rabbit anti-SUMO-2/3 antibody (dilution 1:1000, Cell Signaling Technology, Cat# 4971), respectively.

## Native BrdU assay

To detect nascent ssDNA, native BrdU assay were performed as described[52]. Briefly, U2OS cells were pulse-labeled with 10 μM BrdU (Sigma-Aldrich) for 25 min prior to treatment with 4 mM HU for 2 h.

After washing with PBS, cells were permeabilized with 0.3% Triton X-100 for 5 min at 4 °C and fixed with 3% paraformaldehyde for 10 min at room temperature without any DNA denaturation treatment. Fixed cells were then incubated with mouse anti-BrdU antibody (dilution 1:20, BD Biosciences, Cat# 347580) for 1 h at room temperature, followed by secondary antibody (anti-mouse Alexa Fluor 488 (dilution 1:1000, Thermo Fischer Scientific, Cat# A28175)) and counterstained with DAPI to visualize nuclear DNA. Images were acquired using a Nikon SoRa super-resolution spinning disk microscope equipped with a Plan Fluor 60x oil objective lens and a camera (CoolSNAP HQ2; Photometrics) and analyzed with ImageJ (Fiji, 2020).

## Statistics and reproducibility

Each experiment was repeated three times independently with similar results for all Figs. 1c, 2b, c, 4g, 5a, e, f; Supplementary Figs. 1a–e, 2a, 2b, 3a, 4a, 5d, 6a–d, 7b–e. Statistical analyses were carried out using GraphPad prism 8.0 software. Test methods were described in each figure legend. $*p < 0.05$, $**p < 0.01$, $***p < 0.001$, $****p < 0.0001$ was considered significant and ns=not significant.

## Reporting summary

Further information on research design is available in the Nature Portfolio Reporting Summary linked to this article.

## Data availability

The datasets generated during the current study are available from the corresponding author on request. Source data are provided with this paper.

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

## Acknowledgements

We thank microscopy core facility, Center for Cancer Research (CCR), NCI, NIH for imaging. We are grateful to DNA replication group, DTB, NCI, NIH for their technical assistance. We are also thankful to Professor Lee Zou (Harvard Medical School, Massachusetts General Hospital Cancer Center) for providing EGFP tagged human SPRTN WT and SPRTN ΔUBZ plasmid. This study was supported by the Center for Cancer Research, the Intramural Program of the National Cancer Institute, National Institutes of Health, Bethesda, Maryland 20892 (Grant Z01 BC Z01 BC 006161-17 and Grant Z01 BC 006150-19 to Y.P.).

## Author contributions

Conceptualization: L.K.S., Y.P; Methodology: X.Y., S.S., Y.S., U.J.; Supervision: Y.P.; Writing-original draft: L.K.S.; Writing-review and editing: S.S., S.-y.N.H., Y.P.

## Funding

## Competing interests

The authors declare no competing interests.
