## [Peer Review File · Nature Communications]

Replication-associated formation and repair of human topoisomerase III α cleavage complexesREVIEWER COMMENTS

Reviewer #1 (Remarks to the Author):

The authors performed an excellent work, demonstrated a role of Topoisomerase IIIa (TOP3A) in DNA replication, and identified the repair mechanism when TOP3A fails to complete catalysis. Previous biochemical studies demonstrate that like TOP1, TOP3A cuts and reseals one strand of duplex DNA, and the catalysis transiently generates a TOP-DNA crosslink. While the anti-cancer agent, topotecan stabilizes TOP1-DNA crosslinks, generates lethal DNA double-strand breaks (DSBs) during DNA replication, and kills cancer cells, no inhibitor against TOP3A or TOP3B is available. The known function of TOP3 is to resolve homologous DNA recombination intermediates by physically associating with the Bloom DNA helicase (Blm). The loss of TOP3A, but not the loss of Blm, leads to severe genome instability and lethality, suggesting that TOP3A has an additional unknown function essential for genome maintenance. This additional function remains unknown because the lethality of TOP3A-deficient cells hinders the identification of the cause of genomic instability and no specific inhibitor is available.

In this study, the authors unveiled TOP3A biology in DNA replication using a "self-trapping" TOP3A mutant, the same approach with which they characterized TOP3B in their recent Cell Reports paper. The authors found that a point mutation (R321W) of TOP3A has the same effect as the effect of topotecan (anti-cancer TOP1 inhibitor) on TOP1. R321W stabilizes a catalytic intermediate (stabilized TOP3A(R321W)-DNA cleavage complex, hereafter call TOP3Acc) otherwise it is hardly detectable. Thus, the TOP3A-R321W mutant opens the door to monitor where and when TOP3A actively catalyzes DNA, as topotecan and another TOP1 inhibitor, camptothecin have greatly contributed to the understanding of the essential role played by TOP1 in DNA replication and transcription. The authors showed that TOP3Accs form during DNA replication according to iPOND (isolation of protein on nascent DNA) assays (Fig. 2D) and DNA fiber analyses (Fig. 3A to E). The authors revealed the three repair pathways of TOP3Accs, including SPRTN (Fig. 5), TDP2 (Fig. 6), and Mre11 which works together with ATM and CtIP (Fig. 6). The Role of TOP3A in DNA replication are still unclear, but overall this is an elegant timely study elucidating TOP3A actions in the cell with the TOP3Acc as a readout, uncovering the association of TOP3A with DNA replication and moreover identifying repair pathways for TOP3Acc caused by "abortive" catalysis of TOP3A.

In conclusion, the reviewer strongly recommends the publication of this manuscript in Nature Communications if the authors appropriately revise it.

Specific comments:

1. Fig.1D. TOP3(R364W)-FLAG signals are seen outside the DAPI-stained nucleus. Do they represent TOP3 trapped mitochondria DNA? The authors need to make sure that TOP3(R364W)-FLAG signals (Figs. 1C, 1D, 2C, and 2D) come from genomic DNA but not from mitochondria DNA.
2. Fig.1C and 1D. Why do TOP3-FLAG subnuclear foci are seen in Fig. 1D but not in slot blots in Fig. 1C? Do subnuclear foci reflect non-trapped intact TOP3 accumulating at replication factory? The dynamic range of the slot blot seems to be very limited. To confirm no trapped wild-type TOP3, the authors need to load a larger amount of genomic DNA (such as 20 ug) derived from TOP3-FLAG expressing cells. Please also see the 5th comment below.
3. Fig. 2D in Page 4, paragraph 6, "a detectable fraction of TOP3A remained after thymidine chase (Fig. 2D, lane 9), consistent with the trapping of TOP3Accs by the TOP3A-R364W mutant." The authors should show that the effect of R364W mutation on TOP3 trap in the chase time is significantly larger than the effect of wild-type TOP3 overexpression by showing a histogram. The reviewer requires statistical analysis because TOP3A in NT also remained detectable in lane 3 (the signal ratio between lanes 2 and 3 in Fig. 2D is similar to the ratio between lanes 8 and 9). PCNA and CDC45 also

remained detectable in lane 9, indicating that signal in lane 9 are not necessarily from DPCs and does not support the presence of TOP3Accs or "the trapping of TOP3Accs by the TOP3A-R364W mutant". The authors need to repeat the experiment and show statistical significance in the accumulation of TOP3A-R364W in the chase time.

4. In Fig. 3G is to show the overexpression of mutant TOP3 is likely to delay the progression of S and G2 phases. However, the cell cycle analysis (Fig. 3G) does not show the length of each cell cycle phase. To show a delay in the progression of S and G2 phases, the authors have to show the cell cycle progression of cells expressing wild-type TOP3 and mutant TOP3 after the cell synchronization by the double thymidine block. In addition, the authors need to show the % of cells in the early, middle, and late S phases, because the overexpression of mutant TOP3 may cause the accumulation of cells in the early S phase.

5. Fig. 5B shows that the R364W mutation caused the significant TOP3 trap in SPRTN-proficient cells while wild-type and R364W mutant TOP3s were trapped to a similar extent in SPRTN-depleted cells. The data imply that intact TOP3 is also trapped and SPRTN removes it more efficiently than R364W mutant rather than "These results demonstrate that SPRTN has an important role in the removal of TOP3Accs" (line 12 on page 7). The authors need to confirm whether intact TOP3 is frequently trapped.

6. Fig. 6H showed that TOP3A-R364Wccs were more enriched in G2/M than S in wild-type cells, which result disagrees with the other results (Figs. 2B, 6F, and 6G). Please explain this discrepancy or repeat the experiment of Fig. 6H.

Minor:

Page 2, paragraph 2, the third line. "SSB"
Please spell out SSB.

Page 2, paragraph 2, the fourth line. "TOPA3cc"
Typo. Change "TOPA3cc" into "TOP3Acc."

Page 4, paragraph 2, the 7th line. "similarly expressed (~20-fold)"
Please explain "~20-fold" relative to what.

Page 4, paragraph 4, "TOP3Accs were detected mostly in S-phase (~80%) and less (~20%) in G2/M (Fig. 2B)"

The "%" is confusing. Does this sentence mean "The number of TOP3Accs per S-phase cell was four times larger than that of TOP3Accs per G2/M-phase cell"?

Fig. 2B shows the cell-cycle-phase dependency of trapped TOP3Acc formation and repair, and the authors have to show the cell cycle progression after the cell synchronization by double thymidine block in supplementary figure.

Page 5, paragraph 2, "Binning forks as a function of their velocity showed a leftward shift in cells expression R364W-TOP3A, indicative of a significant reduction of replication fork velocities in response to TOP3A-R364Wccs."

The R364W mutant graph may show bimodal distribution, not a leftward shift. It doesn't represent global reduction in replication speed because replication speed remains high for many replication forks in the cell expressing R364W. Rather, R364W expression induced a significant fraction of low-speed forks, possibly stalled by TOP3A-R364Wccs in a stochastic manner as discussed in the next paragraph on unidirectional and asymmetrical forks.

Page 5, paragraph 3, "a three-fold increase"
The sentence seems incomplete.

Fig. 4B. TOP3-FLAG and RAD51 signals are seen outside the DAPI stained nucleus in TOP3A-R364W.
Please show a better image.

Page 7, paragraph 3, "SPRTN interacts with TOP3A through its catalytic and UBZ regulatory domains."
The co-IP experiment may not suggest direct SPRTN-TOP3A interaction but rather indirect association through chromatin or the replisome complex. Unlike in Fig. 5F, DNA was not digested so that immunoprecipitated TOP3A signal was not from TOP3Accs but free TOP3A. There is still a possibility that SPRTN binds free TOP3A as a degradation target before TOP3A forms an abortive TOP3Acc. However, a simpler interpretation of the data may be that SPRTN and TOP3A (indirectly) interact not because SPRTN targets TOP3A for degradation but because they are both associated with chromatin during replication possibly at stalled replication forks.

Page 7, paragraph 8, "SDS-PAGE and immunoblotting with anti-ubiquitin antibody showed a marked increase in cellular ubiquitination of TOP3Accs"
Precipitates from the RADAR assay should be crude and not pure TOP3Accs. Ubiquitination levels in TOP3Accs cannot be assessed with data shown in Fig. 5F.

Page 8, paragraph 8, "We also found t an additive effect"
Remove "t".

Page 8, paragraph 9, "ATM-CtIP-MRE11 during G2/M"
In Fig. 4E, the authors discussed a possible involvement of ATM in repair of TOP3Acc during S phase. In Fig. 6, the authors concluded that ATM acts in G2/M rather than S phase. Please reconcile this apparent contradiction between interpretations of Fig 4E and 6 results.

Page 11, paragraph 1, "comparable levels of TOP3Accs were observed in siSPRTN, TDP2-/- and siSPRTN/TDP2-/- cells"
This statement is incorrect for TOP3A-WT expressing cells (Fig. 6A). Please discuss the difference in TDP2 dependency between cells expressing TOP3A WT and R364W.

Page 12, paragraph 3, "Western blotting"
Add information on antibodies used in Fig. 4E.

Page 22, Fig. 1B
Add citation. Double-check if the model represents the full length (1-1001) of TOP3A because its C-terminus region seems missing in the figure.

Page 22, Fig. 1D
In Fig. 1D, 2C, and 4B, some foci were located outside of DAPI-positive regions. Please explain the signal in the text or the legend. Are they mitochondrial signal?

Page 23, Fig. 2F
Explain error bars (SD?) in Fig 2F and other figures for RADAR assays.

Page 24, Fig. 3B
Fix x axis values. Median values appear incorrect.

Page 24, Fig. 3D
Please explain x and y axis values and provide definition of asymmetric forks (>1.5-fold difference between left and right replication speeds?).

Page 25, Fig. 4C

Please explain the quantification method in more detail (fraction of TOP3A foci associated with RAD51 foci seem higher than 12%).

Page 25, Fig. 4D

Please explain the quantification method in more detail (Were intensities of γ H2AX measured per nucleus, not per γ H2AX focus? Representative image may help understand the quantification).

Page 25, Fig. 4G

Plot the data as in Fig. 3E because the current graph implies that LC, UFB and CB were mutually exclusive and never present in the same cell (this is not the case as shown in Fig. 4F).

Reviewer #2 (Remarks to the Author):

Saha et al investigate the formation and repair of a self-trapping mutant of TOP3A, which results in stabilized covalent DNA-TOP3A cleavage complexes (TOP3Accs). TOP3A-trapping triggers genome instability, as has been described by the same team for a TOP3B trapping mutant (Saha et al, Cell Rep, 2020). Depletion of the metalloprotease SPRTN results in accumulation of trapped TOP3A, which suggests that SPRTN degrades TOP3Accs for repair. Epistasis analysis with MRE11 and TDP2 suggests that SPRTN acts upstream of TDP2, while MRE11 acts in parallel.

The data presented is overall consistent with the interpretations and with previous publications showing that DNA-protein crosslink cause genome instability and cell cycle and replication defects (e.g. Vaz et al, Mol Cell, 2016). I was surprised that the results are not put in any relation to previous observations by the authors which showed that TOP3Bccs are modified by the TRIM41 E3 ligase and subsequently degraded by the proteasome. Why is TRIM41 not targeting TOP3Accs? Was this tested? Moreover, the authors recently showed that TOP1- and TOP2ccs are SUMOylated, which primes ubiquitylation and proteasomal degradation. Has TOP3Acc-SUMOylation been investigated? Is it important for ubiquitylation? Given that the proteasome was shown to act redundantly with SPRTN during replication-coupled DPC repair (Larsen et al Mol Cell 2019), why was the effect of proteasomal inhibition not tested? In addition to these many gaps, crucial control experiments are missing to ensure that the proposed interpretations of several key observations are correct. Therefore, I am not persuaded that the manuscript offers the conceptual advancement and thoroughness which would warrant publication in Nature Comms.

Major points

The entire manuscript relies on strong overexpression of a mutant protein by transient transfection. It is not surprising that this results in replication defects and genome stability.

What is the effect of physiological relevant levels of TOP3Accs on cellular physiology?

TOP3A is an essential gene with key functions in maintenance of genome stability. Therefore, can the authors exclude that the observed phenotypes (e.g. increased chromatin bridges, checkpoint activation, etc.) are indeed a consequence of TOP3Acc formation and not of dominant-negative effect caused by out-competing endogenous TOP3A from the BTR complex?

The authors show that TOP3Accs form preferentially in S-phase. At the same time, SPRTN depletion causes a cellular arrest in late-S/G2 (Maskey et al, Nat Comms, 2017). Can it be excluded that the accumulation of TOP3Accs upon knockdown of SPRTN is simply a consequence of the associated cell cycle arrest (and not a result of reduced repair)? Similarly, I do not understand how cells can be efficiently synchronized by a double thymidine block, if they express the TOP3A mutant, which is claimed to block replication forks. Can you provide flow cytometry data showing that the synchronization is comparable between cells expressing TOP3A WT vs R364W (Fig 2D)?

The epistasis between TDP2 and SPRTN and the synthetic relationship between TDP2 and MRE11 needs to be confirmed by viability assays. Both claims rely currently only on a single experiment, which could be confounded by many things (e.g. difference in cell cycle phase, etc).

Minor points

The authors observe that a catalytic inactive SPRTN mutant (E112A) fails to bind TOP3A and conclude that the catalytic domain is important for their interaction. I cannot follow this interpretation. This mutant has an intact catalytic domain and should be fully proficient to interact with the substrate (which is independent of the catalytic residue). In general, catalytic inactive enzyme variants bind typically stronger to their substrates because they cannot process them.

siRNA is used in the study to deplete SPRTN. The authors have generated SPRTN KO cells in of their recent papers (Saha et al., NAR, 2021). SPRTN's role in TOP3Acc repair and the epistasis with TDP2 should be confirmed in these cells.

Figure C: What is the reason for the non-nuclear TOP3Acc signals? Are these mitochondrial TOP3Accs?

Reviewer #3 (Remarks to the Author):

In this manuscript Saha et al. extend their previous work with a self-trapping TOP3B mutant to the TOP3A paralog, and use this mutation to explore the roles of TOP3A in replication and the mechanisms of DNA-protein crosslink (DPC) repair. Overall, the experiments are well designed and the results are sound and of interest to researchers in the field, but, as they stand, we do not find them to significantly extend our knowledge on the functions of TOP3B or on DPC repair mechanisms.

Regarding the function on replication, the proposal of TOP3A acting behind the fork to remove precatenanes is interesting, but it is neither sufficiently proven in the present manuscript, nor a novel idea (see Bakx 2022 Nature Communications, which the authors fail to cite). In this sense, we think that the results presented, in their current form, are fully compatible with known functions of TOP3A in dissolving recombination intermediates, which would be, as expected, associated to the replication fork. Compelling evidence should be provided to demonstrate a recombination-independent function of TOP3A at the fork, as well as providing solid mechanistic insights into what this function may be.

Regarding DPC repair, the results are mostly in line with what is already known from repair of TOP2ccs. In addition, the model presented, in which TDP2 operates after TOP3A protein degradation, is somewhat incompatible with the accumulation of fully detectable TOP3Accs in the absence of TDP2, as the protein should be already proteolysed. This would only occur if the antibody detects the region surrounding the covalently bound catalytic site. Although this is formally possible, it would need to be experimentally proven. It is also puzzling that while TDP2 and SPRTN deficiency lead to similar accumulation of TOP3A-R363W cc accumulation, they have a very different effect on the accumulation of wt protein ccs. Are the two proteins recognized and degraded equally? All the experiments with MRE11si might be affected by the replicative problems that its absence may cause. The authors should consider this, and, at least, provide clear evidence that that the effects observed are not due to changes in the replicative status and/or cell cycle stage of the cells. Finally, in order to claim a function of ATM in this process, the authors should include ATM deleted/depleted cells.

The last major point is regarding the appropriateness of a model exclusively based on protein over-expression. In Figure 2D, it is clear that TOP3A-overexpressing cells present a highly increased recruitment to the fork. It is therefore important to completely rule out that the effects observed are artefactually derived from the high over-expression conditions, and demonstrate that they truly reflect physiological functions of TOP3A at the fork. At least some of the key experiments should be reproduced with physiological levels of expression of the TOP3A-R363W mutant.

Additional comments:

- What is the non-nuclear signal in Fig 1D?
- Asymmetric chromatin fibers could also represent broken forks.
- How was the clonogenic experiments in Fig A quantified? In M&M it indicates that the results are related to NT control, but this is not compatible with the figure.
- The increase in Rad51 foci and not only % of colocalization should be shown in Fig 4B-C.

Reviewer #1 (Remarks to the Author):

The authors performed an excellent work, demonstrated a role of Topoisomerase IIIa (TOP3A) in DNA replication, and identified the repair mechanism when TOP3A fails to complete catalysis. Previous biochemical studies demonstrate that like TOP1, TOP3A cuts and reseals one strand of duplex DNA, and the catalysis transiently generates a TOP-DNA crosslink. While the anti-cancer agent, topotecan stabilizes TOP1-DNA crosslinks, generates lethal DNA double-strand breaks (DSBs) during DNA replication, and kills cancer cells, no inhibitor against TOP3A or TOP3B is available. The known function of TOP3 is to resolve homologous DNA recombination intermediates by physically associating with the Bloom DNA helicase (Blm). The loss of TOP3A, but not the loss of Blm, leads to severe genome instability and lethality, suggesting that TOP3A has an additional unknown function essential for genome maintenance. This additional function remains unknown because the lethality of TOP3A-deficient cells hinders the identification of the cause of genomic instability and no specific inhibitor is available.

In this study, the authors unveiled TOP3A biology in DNA replication using a "self-trapping" TOP3A mutant, the same approach with which they characterized TOP3B in their recent Cell Reports paper. The authors found that a point mutation (R364W) of TOP3A has the same effect as the effect of topotecan (anti-cancer TOP1 inhibitor) on TOP1. R364W stabilizes a catalytic intermediate (stabilized TOP3A(R364W)-DNA cleavage complex, hereafter called TOP3Acc) otherwise it is hardly detectable. Thus, the TOP3A-R364W mutant opens the door to monitor where and when TOP3A actively catalyzes DNA, as topotecan and another TOP1 inhibitor, camptothecin have greatly contributed to the understanding of the essential role played by TOP1 in DNA replication and transcription. The authors showed that TOP3Accs form during DNA replication according to iPOND (isolation of protein on nascent DNA) assays (Fig. 2D) and DNA fiber analyses (Fig. 3A to E). The authors revealed the three repair pathways of TOP3Accs, including SPRTN (Fig. 5), TDP2 (Fig. 6), and Mre11 which works together with ATM and CtIP (Fig. 6). The Role of TOP3A in DNA replication are still unclear, but overall, this is an elegant timely study elucidating TOP3A actions in the cell with the TOP3Acc as a readout, uncovering the association of TOP3A with DNA replication and moreover identifying repair pathways for TOP3Acc caused by "abortive" catalysis of TOP3A.

In conclusion, the reviewer strongly recommends the publication of this manuscript in Nature Communications if the authors appropriately revise it.

Response: Thank you for finding our study timely and of broad interest in topoisomerase and fundamental cell biology. We appreciate your constructive comments. As suggested, we have addressed your specific comments point-by-point below and revised our manuscript accordingly.

Specific comments:

1. Fig.1D. TOP3(R364W)-FLAG signals are seen outside the DAPI-stained nucleus. Do they represent TOP3 trapped mitochondria DNA? The authors need to make sure that TOP3(R364W)-FLAG signals (Figs. 1C, 1D, 2C, and 2D) come from genomic DNA but not from mitochondria DNA.

Response: Thank you for the comment. Indeed, as indicated in the manuscript, TOP3A is present in both the nucleus and mitochondria. In this report, we have studied selectively nuclear TOP3A, not mitochondrial TOP3A. Using immunofluorescence microscopy in Fig.1D, we have seen the presence of TOP3A-FLAG signals both inside and outside of DAPI-stained nuclei. TOP3A-R364W-FLAG signals that were seen outside the DAPI stained nucleus are indeed mitochondrial TOP3A. We confirmed the presence of these mitochondrial TOP3A by co-staining cells with anti-TOMM20 (mitochondrial surface marker) antibody and anti-FLAG antibody. We have added the data in Supplementary Fig. 1D of the revised manuscript, and added this point in the Result section of the revised manuscript.

For Fig.1C, RADAR assay uses whole cell lysate with relatively small amount of starting material, usually approximately 1 million cells. Therefore, we assume that there is negligible amount of mitochondrial DPC captured from 1 million cells, and that the signal is primarily from nuclear TOP3A-DPCs.

2. Fig.1C and 1D. Why do TOP3A-FLAG subnuclear foci are seen in Fig. 1D but not in slot blots in Fig. 1C? Do subnuclear foci reflect non-trapped intact TOP3 accumulating at replication factory? The dynamic range of the slot blot seems to be very limited. To confirm no trapped wild-type TOP3, the authors need to load a larger amount of genomic DNA (such as 20 ug) derived from TOP3A-FLAG expressing cells. Please also see the 5th comment below.

Response: Thank you. The experimental conditions between Fig.1C and 1D are different. Fig. 1C shows covalent TOP3A cleavage complexes (TOP3Accs) detected by the RADAR assay, which isolates DNA with the covalently bound protein adducts. By contrast, in Fig. 1D, the detection of TOP3A foci does not require the covalent linkage of TOP3A to DNA; it detects both non-trapped and trapped TOP3A visualized by immunofluorescence microscopy (IF) after pre-extracting cells with mild non-ionic detergent (0.2% Triton X-100). In agreement with the accumulation of TOP3A at replication factories, Fig. 2C provides evidence for the colocalization of TOP3A and replication foci measured as CDC45 and RPA foci. Although we cannot exclude that wild-type TOP3A overexpression leads to TOP3Accs, even after loading large amount of DNA such as 20 µg, trapped TOP3A was not detectable in cells expressing WT TOP3A-FLAG.

3. Fig. 2D in Page 4, paragraph 6, "a detectable fraction of TOP3A remained after thymidine chase (Fig. 2D, lane 9), consistent with the trapping of TOP3Accs by the TOP3A-R364W mutant." The authors should show that the effect of R364W mutation on TOP3 trap in the chase time is significantly larger than the effect of wild-type TOP3A overexpression by showing a histogram. The reviewer requires statistical analysis because TOP3A in NT also remained detectable in lane 3 (the signal ratio between lanes 2 and 3 in Fig. 2D is similar to the ratio between lanes 8 and 9). PCNA and CDC45 also remained detectable in lane 9, indicating that signal in lane 9 are not necessarily from DPCs and does not support the presence of TOP3Accs or "the trapping of TOP3Accs by the TOP3A-R364W mutant". The authors need to repeat the experiment and show statistical significance in the accumulation of TOP3A-R364W in the chase time.

Response: As suggested, we confirmed our finding that a small but significant fraction of TOP3A remained after thymidine chase in cells expressing TOP3A-R364W. These experiments were quantified and signal in TOP3A R364W was compared to cells expressing TOP3A-WT. We observed approximately three-fold increased TOP3A signal in TOP3A-R364W-expressing cells in comparison with TOP3A-WT cells. We have added this quantification with statistical significance in Fig. 2E of the revised manuscript. We normalized the chase signal with click signal of TOP3A as well as H3 signal. Our explanation by comparing signal between lane 6 and lane 9, where TOP3A chase signal in TOP3A-WT expressing cells was hardly detectable but TOP3A-R364W expressing cells displayed a significant remaining fraction of TOP3A signal, which we interpret as resulting from slower replication due to the trapping of TOP3Accs. We have modified the text in the revised manuscript as “In addition, in cells expressing TOP3A-R364W, detectable fraction of TOP3A remained after thymidine chase when compared to TOP3A-WT expressing cells, which likely results from trapped TOP3Accs (Fig. 2D, compare lanes 6 and 9)”. Thank you.

4. In Fig. 3G is to show the overexpression of mutant TOP3 is likely to delay the progression of S and G2 phases. However, the cell cycle analysis (Fig. 3G) does not show the length of each cell cycle phase. To show a delay in the progression of S and G2 phases, the authors have to show the cell cycle progression of cells expressing wild-type TOP3 and mutant TOP3 after the cell synchronization by the double thymidine block. In addition, the authors need to show the % of cells in the early, middle, and late S phases, because the overexpression of mutant TOP3 may cause the accumulation of cells in the early S phase.

Response: Thank you. As suggested, we have included the data with cell synchronization by double thymidine block, which are presented in Supplementary Fig. 2D of the revised manuscript. We have also updated Fig. 3G and shown the percentage of cells in the early, middle, and late S phases in the revised manuscript. In Fig. 3H, we show the quantitation data of 3G. The Result section of the revised manuscript has been modified accordingly. Together these data demonstrate that the toxic TOP3A slows S-phase progression with accumulation of cells in the early S-phase.

5. Fig. 5B shows that the R364W mutation caused the significant TOP3 trap in SPRTN-proficient cells while wild-type and R364W mutant TOP3s were trapped to a similar extent in SPRTN-depleted cells. The data imply that intact TOP3 is also trapped and SPRTN removes it more efficiently than R364W mutant rather than “These results demonstrate that SPRTN has an important role in the removal of TOP3Accs” (line 12 on page 7). The authors need to confirm whether intact TOP3 is frequently trapped.

Response: Thank you. We believe that the reviewer means intact TOP3A as *wild-type* TOP3A. We did not see any detectable fraction of TOP3Accs in endogenous level as well as TOP3A-WT expressing control (*wild-type*) cells. The detectable level of TOP3Accs in SPRTN-deficient cells after expressing TOP3A-WT plasmid suggest SPRTN might be major player to degrade TOP3A that is covalently bound to DNA in S-phase, given our observations that TOP3A forms predominantly in S-phase. Abortive trapping of topoisomerases could happen due to many

endogenous causes in cells, but its detection is still beyond the limit by currently established methods such as RADAR and ICE assays, possibly due to the small amount as well as faster turnover dynamics of TOPccs. Hence, our results of no detectable level of WT TOP3A are not at odd with the prior biology of the endogenous topoisomerase cleavage complexes for the more extensively studied TOP1 and TOP2.

6. Fig. 6H showed that TOP3A-R364Wccs were more enriched in G2/M than S in wild-type cells, which result disagrees with the other results (Figs. 2B, 6F, and 6G). Please explain this discrepancy or repeat the experiment of Fig. 6H.

Response: We apologize for the editing mistake for the blot in Fig. 6H in the original manuscript. TOP3Accs induced by TOP3A-R364W were consistently enriched in S- compared to G2/M-phase cells. We have replaced the blot in the revised manuscript. The figure is now included in Fig. 7 as panel C.

Minor:

Page 2, paragraph 2, the third line. “SSB”
Please spell out SSB.

Response: We have spelled out SSB as ‘single-strand breaks’ in the revised manuscript.

Page 2, paragraph 2, the fourth line. "TOPA3cc"
Typo. Change "TOPA3cc" into "TOP3Acc."

Response: Thank you. We have corrected the typo in the revised manuscript.

Page 4, paragraph 2, the 7th line. "similarly expressed (~20-fold)"
Please explain “~20-fold” relative to what.

Response: We have revised the text as “~20-fold relative to mock-transfected”

Page 4, paragraph 4, "TOP3Accs were detected mostly in S-phase (~80%) and less (~20%) in G2/M (Fig. 2B)" The “%” is confusing. Does this sentence mean “The number of TOP3Accs per S-phase cell was four times larger than that of TOP3Accs per G2/M-phase cell”?

Response: Yes, this sentence means “The number of TOP3Accs per S-phase cell was four times larger than that of TOP3Accs per G2/M-phase cell”. To avoid any confusion, we have revised the sentence as “The level of TOP3Accs in S-phase cells was four-fold higher than that of TOP3Accs in G2/M-phase cell” in the revised manuscript.

Fig. 2B shows the cell-cycle-phase dependency of trapped TOP3Acc formation and repair, and the authors have to show the cell cycle progression after the cell synchronization by double thymidine block in supplementary figure.

Response: As suggested, we have included the flow cytometry data to show the cell cycle progression after the cell synchronization by double thymidine block in Supplementary Fig. 2D.

Page 5, paragraph 2, "Binning forks as a function of their velocity showed a leftward shift in cells expression R364W-TOP3A, indicative of a significant reduction of replication fork velocities in response to TOP3A-R364Wccs."The R364W mutant graph may show bimodal distribution, not a leftward shift. It doesn't represent global reduction in replication speed because replication speed remains high for many replication forks in the cell expressing R364W. Rather, R364W expression induced a significant fraction of low-speed forks, possibly stalled by TOP3A-R364Wccs in a stochastic manner as discussed in the next paragraph on unidirectional and asymmetrical forks.

Response: Thank you. We agree with the point raised by the reviewer that 'R364W expression induced a significant fraction of low-speed forks, possibly stalled by TOP3A-R364Wccs in a stochastic manner'. In the revised manuscript, we have removed this part in the sentence: 'Binning forks as a function of their velocity showed a leftward shift in cells expression R364W-TOP3A'.

Page 5, paragraph 3, "a three-fold increase"
The sentence seems incomplete.

Response: Thank you for noting this editorial error; we have edited the text to make the sentence complete as "a 2-3-fold increase in comparison with TOP3A-WT-expressing cells" in the revised manuscript.

Fig. 4B. TOP3-FLAG and RAD51 signals are seen outside the DAPI stained nucleus in TOP3A-R364W. Please show a better image.

Response: As suggested, a better image is provided in Fig. 4B of the revised manuscript.

Page 7, paragraph 3, "SPRTN interacts with TOP3A through its catalytic and UBZ regulatory domains."The co-IP experiment may not suggest direct SPRTN-TOP3A interaction but rather indirect association. Unlike in Fig. 5F, DNA was not digested so that immunoprecipitated TOP3A signal was not from TOP3Accs but free TOP3A. There is still a possibility that SPRTN binds free TOP3A as a degradation target before TOP3A forms an abortive TOP3Acc. However, a simpler interpretation of the data may be that SPRTN and TOP3A (indirectly) interact not because SPRTN targets TOP3A for degradation but because they are both associated with chromatin during replication possibly at stalled replication forks.

Response: In this study, we did not wish to imply that SPRTN and TOP3A interact with each other directly. In principle, it is plausible that they are indirectly associated through chromatin or the replisome complex. Of note, we also digested the nucleic acids in the lysates with benzonase during co-IP experiment as mentioned in Material and Methods section. Given that a main and well-studied function of SPRTN is to act as a replication-associated DPC protease and the observed increased level of TOP3Accs in SPRTN-deficient cells in this study led us to propose that SPRTN mainly target TOP3Accs. In our experimental setting, the aim was to examine the binding of SPRTN with TOP3A that could be TOP3Accs and free TOP3A both after isolating chromatin fraction from cells. However, we can't fully exclude the possibility that SPRTN binds

free TOP3A as well, as reviewer mentioned. Additionally, as we did the IP experiment in TOP3A-R364W expressing cells and the observations of reduced interaction between TOP3A and mutant SPRTN might be suggestive of mainly TOP3Accs and SPRTN interaction.

Page 7, paragraph 8, "SDS-PAGE and immunoblotting with anti-ubiquitin antibody showed a marked increase in cellular ubiquitination of TOP3Accs". Precipitates from the RADAR assay should be crude and not pure TOP3Accs. Ubiquitination levels in TOP3Accs cannot be assessed with data shown in Fig. 5F.

Response: For Fig. 5F, we performed our established method, DUST assay which was verified by our previous studies to detect ubiquitylated and SUMOylated TOPccs [Sun Y. et al., *Sci Adv.* 2020 (PMID: 33188014) and Saha S. et al., *Cell Rep.* 2020 (PMID: 33378676)]. Also, it is well established that the RADAR assay uses chaotropic reagents that removes non-covalently bound proteins. Therefore, any other polyubiquitylated proteins potentially associated with TOP3Accs would have to be covalently attached to DNA. This point has been clarified in our revised manuscript.

Page 8, paragraph 8, "We also found t an additive effect" Remove "t".

Response: Sorry for the typo. We removed "t" in the revised manuscript.

Page 8, paragraph 9, "ATM-CtIP-MRE11 during G2/M"

In Fig. 4E, the authors discussed a possible involvement of ATM in repair of TOP3Acc during S phase. In Fig. 6, the authors concluded that ATM acts in G2/M rather than S phase. Please reconcile this apparent contradiction between interpretations of Fig 4E and 6 results.

Response: In Fig. 4E (Fig. 4G in the revised manuscript), we performed Western blot using whole cells extracts from asynchronous cell populations. We interpreted the data of ATM activation (phospho-ATM) as a consequence of DNA damage response without implying the selective involvement of ATM in the repair of TOP3Accs during S phase. The results in Fig. 6 suggest that the involvement of ATM in TOP3Accs repair independently of TDP2 (Fig. 6F in the revised manuscript). However, we provide evidence that ATM is epistatic with CtIP (Fig. 6G in the revised manuscript) and that MRE11 acts in parallel with TDP2 (Fig.6D and E in the revised manuscript). Based on these findings, we propose that the ATM-CtIP-MRE11 axis may preferentially repair TOP3Accs during G2/M (Fig. 7D of the revised manuscript).

Page 11, paragraph 1, "comparable levels of TOP3Accs were observed in siSPRTN, TDP2^{-/-} and siSPRTN/TDP2^{-/-} cells" This statement is incorrect for TOP3A-WT expressing cells (Fig. 6A). Please discuss the difference in TDP2 dependency between cells expressing TOP3A WT and R364W.

Response: Thank you. We have edited the text as "comparable levels of TOP3Accs were observed in siSPRTN, TDP2^{-/-} and siSPRTN/TDP2^{-/-} cells after TOP3A-R364W transfection" in the revised manuscript. Based on our own experience and published studies by other group, endogenous TOPccs (TOP1ccs, TOP2ccs and TOP3Bccs) are not detectable by ICE or RADAR assays in cultured mammalian cells after complete depletion of TDP genes involved in their excision. Their detection requires treatment with topoisomerase poisons (camptothecins for

TOP1 or etoposide for TOP2). However, we could not detect TOP3Accs in TOP3A-WT-expressing cells like TOP3Bccs. Hence, our results for TOP3Accs in TDP2-deficient cells are not at odd with the prior biology of the endogenous topoisomerase cleavage complexes for the more extensively studied TOP1 and TOP2.

Page 12, paragraph 3, "Western blotting"
Add information on antibodies used in Fig 4E.

Response: As suggested, we have added the information for the antibodies used in Fig. 4E (Fig. 4G of the revised manuscript) in the Materials and Methods section of the revised manuscript.

Page 22, Fig. 1B
Add citation. Double-check if the model represents the full length (1-1001) of TOP3A because its C-terminus region seems missing in the figure.

Response: The model represented in Fig. 1B includes N-terminus region and catalytic core (aa 1-637), not the full length of TOP3A. The structure of the C-terminus region has not been resolved thus far. As suggested, we have corrected the text and added a citation (Bizard AH et. al., 2019; PMID: 30232008) in the legends of Fig. 1B of the revised manuscript.

Page 22, Fig. 1D
In Fig. 1D, 2C, and 4B, some foci were located outside of DAPI-positive regions. Please explain the signal in the text or the legend. Are they mitochondrial signal?

Response: Indeed, The TOP3A foci located outside of DAPI-positive regions are mitochondrial. We have confirmed this by immunofluorescence after co-staining cells with anti-FLAG and anti-TOMM20 antibody. We have added the data in Supplementary Fig. 1D and corresponding text in the revised manuscript.

Page 23, Fig. 2F
Explain error bars (SD?) in Fig 2F and other figures for RADAR assays.

Response: Yes, the error bars represent SD. This has been clarified in legends of Figs. 2F, 5B, 5D, 6B, 6D-G in the revised manuscript.

Page 24, Fig. 3B
Fix x axis values. Median values appear incorrect.

Response: Thank you. We apologize for this error, and have corrected the median values presented in Fig. 3B of the revised manuscript. We also edited the text in Results section accordingly.

Page 24, Fig. 3D
Please explain x and y axis values and provide definition of asymmetric forks (>1.5-fold difference between left and right replication speeds?).

Response: The values of x and y axis represent replication speed (kb/min) of right and left forks, respectively. Those forks were classified as asymmetric forks when the difference between lengths of left forks and right forks emanating from the same origins was greater than 30%. To clarify this, we have added the text in figure legends of Fig. 3D of the revised manuscript as reviewer suggested.

Page 25, Fig. 4C

Please explain the quantification method in more detail (fraction of TOP3A foci associated with RAD51 foci seem higher than 12%).

Response: We quantified the percentage of cells with co-localization of TOP3A with RAD51 foci in Fig. 4C (Fig. 4D of the revised manuscript). We have clarified the text in the figure legends accordingly in the revised manuscript.

Page 25, Fig. 4D

Please explain the quantification method in more detail (Were intensities of γ H2AX measured per nucleus, not per γ H2AX focus? Representative image may help understand the quantification).

Response: Yes, we measured the intensities of γ H2AX per nucleus. We have included it in the text of legends of Fig. 4D (Fig. 4F of the revised manuscript). As suggested, we also included representative images in Fig. 4E of the revised manuscript.

Page 25, Fig. 4G

Plot the data as in Fig. 3E because the current graph implies that LC, UFB and CB were mutually exclusive and never present in the same cell (this is not the case as shown in Fig. 4F).

Response: Thank you for the comment. We have plotted the data as suggested and updated Fig. 4G (Fig. 4I in the revised manuscript) accordingly.

Reviewer #2 (Remarks to the Author):

Saha et al investigate the formation and repair of a self-trapping mutant of TOP3A, which results in stabilized covalent DNA-TOP3A cleavage complexes (TOP3Accs). TOP3A-trapping triggers genome instability, as has been described by the same team for a TOP3B trapping mutant (Saha et al, Cell Rep, 2020). Depletion of the metalloprotease SPRTN results in accumulation of trapped TOP3A, which suggests that SPRTN degrades TOP3Accs for repair. Epistasis analysis with MRE11 and TDP2 suggests that SPRTN acts upstream of TDP2, while MRE11 acts in parallel.

The data presented is overall consistent with the interpretations and with previous publications showing that DNA-protein crosslink cause genome instability and cell cycle and replication defects (e.g., Vaz et al, Mol Cell, 2016).

I was surprised that the results are not put in any relation to previous observations by the authors which showed that TOP3Bccs are modified by the TRIM41 E3 ligase and subsequently degraded by the proteasome. Why is TRIM41 not targeting TOP3Accs? Was this tested?

Response: Thank you for the suggestion. Indeed, because of the prior connection of TRIM41 with TOP3Bccs (Saha S. et al., Cell Rep. 2020), we investigated the potential connection of TRIM41 with TOP3Acc repair. We found that depletion of TRIM41 had no impact on the level of TOP3Accs (see Supplementary Fig. 6C in the revised manuscript) indicating that TOP3Accs are not targeted by TRIM41. This result is not surprising as TOP3B and TOP3A have very different cellular functions and form different protein complexes. In addition, we have tested the effect of the proteasome in TOP3Accs repair. Treatment of TOP3A-R364W-expressing cells with the proteasome inhibitor, bortezomib (BTZ), caused no changes in the TOP3Accs levels, indicating that TOP3Accs are not degraded by the proteasome. This finding is notable because other TOPccs including TOP1, TOP2A, TOP2B and TOP3Bccs have been shown to be degraded by the proteasome machinery. We have included these new data in Supplementary Fig. 6A, C and D and the Results section of the revised manuscript..

Moreover, the authors recently showed that TOP1- and TOP2ccs are SUMOylated, which primes ubiquitylation and proteasomal degradation. Has TOP3Acc-SUMOylation been investigated? Is it important for ubiquitylation? Given that the proteasome was shown to act redundantly with SPRTN during replication-coupled DPC repair (Larsen et al Mol Cell 2019), why was the effect or proteasomal inhibition not tested?

Response: Thank you for raising this important question. Yes, we have investigated the TOP3Acc-SUMOylation by performing DUST (Detection of Ubiquitylated and SUMOylated TOP-DPCs) assay as described previously by our group (Sun Y. et al., Sci Adv. 2020). Interestingly, we did not observe SUMOylation of TOP3A with antibodies targeting SUMO-1 and SUMO-2/3, suggesting that TOP3Accs are not SUMOylated. Thus, SUMOylation of TOP3A is not required for its ubiquitylation in response to TOP3Accs. As stated above, we have also investigated the role of the proteasome on TOP3Accs repair and found that proteasome inhibition has no detectable effect on TOP3Acc level. We have included these new data in Supplementary Fig. 6B and updated the Results section of the revised manuscript accordingly.

Major points

The entire manuscript relies on strong overexpression of a mutant protein by transient transfection. It is not surprising that this results in replication defects and genome stability. What is the effect of physiological relevant levels of TOP3Accs on cellular physiology?

Response: We agree that our study relies on the overexpression of TOP3A plasmids. Nonetheless, given the observation that TOP3A-WT and TOP3A-R364W expressed in similar level, TOP3A-WT expressing cells displayed no replication defects and genome instability.

These findings exclude the fact that defective phenotypes were originated from the overexpression of protein.

To investigate the effect of physiological relevant levels of TOP3Accs on cellular biology, we performed DNA combing assay after silencing TOP3A in U2OS cells. TOP3A-depleted cells showed reduced replication speed as well as increased new origin firing in comparison with *wild-type* cells. We have included these new data in Supplementary Figure 3 of the revised manuscript. Therefore, we believe that our study suggests that TOP3A is associated with replication during S-phase, and that interfering with the turnover of the cleavage complexes damages replication forks.

TOP3A is an essential gene with key functions in maintenance of genome stability. Therefore, can the authors exclude that the observed phenotypes (e.g. increased chromatin bridges, checkpoint activation, etc.) are indeed a consequence of TOP3Acc formation and not of dominant-negative effect caused by out-competing endogenous TOP3A from the BTR complex?

Response: As we have seen that cells express very similar level of TOP3A when we transfected cells with TOP3A-WT and TOP3A-R364W (Supplementary Fig. 1A-C), we conclude that the observed phenotypes (e.g., increased chromatin bridges, checkpoint activation, etc.) are not consequence of a dominant-negative effect caused by out-competing endogenous TOP3A from the BTR complex. Rather, we interpret our results as a consequence of TOP3Acc formation by mutant TOP3A. Moreover, we have seen coupled upregulation of BLM and RMI1 (components of BTR complex) in cells both in cells overexpressing TOP3A-WT and TOP3A-R364W (Supplementary Fig. 1E in the revised manuscript). These observations rule out the possibility that the observed phenotypes are due to the dominant-negative effect caused by out-competing endogenous TOP3A from the BTR complex.

The authors show that TOP3Accs form preferentially in S-phase. At the same time, SPRTN depletion causes a cellular arrest in late-S/G2 (Maskey et al, Nat Comms, 2017). Can it be excluded that the accumulation of TOP3Accs upon knockdown of SPRTN is simply a consequence of the associated cell cycle arrest (and not a result of reduced repair)? Similarly, I do not understand how cells can be efficiently synchronized by a double thymidine block, if they express the TOP3A mutant, which is claimed to block replication forks. Can you provide flow cytometry data showing that the synchronization is comparable between cells expressing TOP3A WT vs R364W (Fig 2D)?

Response: As requested, we have provided flow cytometry data in the revised manuscript showing that the synchronization is comparable between cells expressing TOP3A WT vs R364W. The data are included in Supplementary Fig. 2D of the revised manuscript. Regarding the cell cycle dependency of TOP3Acc repair, we cannot totally exclude that the effect of SPRTN on cell cycle progression may contribute to additional indirect repair defects. This point has been mentioned in the revised manuscript.

The epistasis between TDP2 and SPRTN and the synthetic relationship between TDP2 and MRE11 needs to be confirmed by viability assays. Both claims rely currently only on a single experiment, which could be confounded by many things (e.g., difference in cell cycle phase, etc).

Response: As suggested, we performed cell viability assays by assessing the colony formation efficiency of indicated genotypes and included these new data in Fig. 6B and E of the revised manuscript. We confirmed that the epistasis between SPRTN and TDP2 (Fig. 6B) and the synergistic relationship between MRE11 and TDP2 (Fig. 6E), which is consistent with the TOP3Accs result in Fig. 6 of the revised manuscript.

Minor points

The authors observe that a catalytic inactive SPRTN mutant (E112A) fails to bind TOP3A and conclude that the catalytic domain is important for their interaction. I cannot follow this interpretation. This mutant has an intact catalytic domain and should be fully proficient to interact with the substrate (which is independent of the catalytic residue). In general, catalytic inactive enzyme variants bind typically stronger to their substrates because they cannot process them.

Response: We performed SPRTN-IP experiments after transfecting cells with toxic TOP3A (TOP3A-R364W) and found that the catalytic-dead SPRTN mutant failed to interact with TOP3A. We agree that this result is also surprising as well as intriguing as the catalytic residue of proteins is not important for the binding to their substrates.

siRNA is used in the study to deplete SPRTN. The authors have generated SPRTN KO cells in of their recent papers (Saha et al., NAR, 2021). SPRTN's role in TOP3Acc repair and the epistasis with TDP2 should be confirmed in these cells.

Response: As suggested, we have examined the levels of TOP3Accs in our SPRTN KO TK6 cells and found that SPRTN KO cells exhibit elevated level of TOP3Accs in comparison with their wild-type counterparts, confirming SPRTN's role in TOP3Accs repair. We also confirmed epistasis of SPRTN with TDP2 in TK6 cells in terms of TOP3Accs repair. We have included these new data in Supplementary Fig. 7A of the revised manuscript.

Figure C: What is the reason for the non-nuclear TOP3Acc signals? Are these mitochondrial TOP3Accs?

Response: Yes, the non-nuclear TOP3Acc signals represents mitochondrial TOP3Accs. The non-nuclear TOP3Acc signal colocalizes with mitochondrial network, which is known in literature and was confirmed by co-staining cells with anti-TOMM20 antibody and anti-FLAG antibody. We have provided the data in Supplementary Fig. 1D of the revised manuscript.

Reviewer #3 (Remarks to the Author):

In this manuscript Saha et al. extend their previous work with a self-trapping TOP3B mutant to the TOP3A paralog, and use this mutation to explore the roles of TOP3A in replication and the mechanisms of DNA-protein crosslink (DPC) repair. Overall, the experiments are well designed and the results are sound and of interest to researchers in the field, but, as they stand, we do not find them to significantly extend our knowledge on the functions of TOP3B or on DPC repair mechanisms.

Response: Thank you for finding our study well designed, the results sound and the study of interest to researchers in the field.

Regarding the function on replication, the proposal of TOP3A acting behind the fork to remove precatenanes is interesting, but it is neither sufficiently proven in the present manuscript, nor a novel idea (see Bakx 2022 Nature Communications, which the authors fail to cite). In this sense, we think that the results presented, in their current form, are fully compatible with known functions of TOP3A in dissolving recombination intermediates, which would be, as expected, associated to the replication fork. Compelling evidence should be provided to demonstrate a recombination-independent function of TOP3A at the fork, as well as providing solid mechanistic insights into what this function may be.

Response: Thank you. We have cited the Bakx et. al., 2022, Nature Communications paper in the Discussion of the revised manuscript. Based on earlier publications and the requirement of single-stranded DNA (ssDNA) for TOP3A binding, we hypothetically proposed the activity of TOP3A in the Okazaki fragment of lagging strands behind replication forks to remove precatenanes. Importantly, our study provides for the first-time direct evidence of TOP3A presence in active replisomes using human cells.

In vivo, it remains challenging to separate recombination and replication as they are intimately linked. To demonstrate a recombination-independent function of TOP3A at the fork, we directly examined DNA replication at single-molecule level by performing DNA combing assays after knocking-down of TOP3A by siRNA in U2OS cells. We found that TOP3A-depleted cells show reduced replication speed as well as increased new origin firing and shorter inter-origin distances. The same phenotype was observed in the R364W-TOP3A-transfected cells. These new data are included in Supplementary Fig. 3 of the revised manuscript. Our study does not focus on recombination independently of TOP3A functions at replication forks.

Replication fork reversal is thought to be a mechanism for the stabilization of forks. To determine whether TOP3A is important for promoting replication fork reversal upon replication stress, we performed native BrdU assay after pulse labelling (25 min) with BrdU (new Supplementary Figure 4A), where BrdU antibody can only recognizes the BrdU labelled nascent ssDNA, not intact replication forks. As shown in Supplementary Figure 4B, in *wild-type* cells, HU treatment caused the induction of BrdU foci that represent ssDNA exposure of nascent DNA strands, indicative of fork reversal. Strikingly, TOP3A depletion by siRNA transfection led to a dramatic reduction in HU-induced ssDNA focus formation (Supplementary Fig. 4B and C).

We also examined whether TOP3A is recruited to stalled replication forks. Accordingly, immunofluorescence microscopy revealed that TOP3A is specifically enriched in dense ssDNA foci (Supplementary Fig. 4D). However, TOP3A-R364W expressing cells showed very similar pattern of co-localization of TOP3A with ssDNA regions as with TOP3A-WT expressing cells. These observations clearly suggest the importance of TOP3A in promoting replication fork reversal *in vivo*. We have included these data in Supplementary Fig. 4 of the revised manuscript and added text accordingly.

Regarding DPC repair, the results are mostly in line with what is already known from repair of TOP2ccs. In addition, the model presented, in which TDP2 operates after TOP3A protein degradation, is somewhat incompatible with the accumulation of fully detectable TOP3Accs in the absence of TDP2, as the protein should be already proteolyzed. This would only occur if the antibody detects the region surrounding the covalently bound catalytic site. Although this is formally possible, it would need to be experimentally proven.

Response: As TOP3A forms 5'-DPCs like TOP2, it is reasonable that some repair machinery might be common between cellular TOP2ccs and TOP3Accs. Yet, TOP3Accs are not SUMOylated but appear to be primarily proteolyzed by SPRTN. This has been included in Supplementary Fig. 6B.

The observed increase of TOP3Accs upon TDP2 depletion confirms that the proteolysis by SPRTN is not sufficient to remove TOP3Accs in the absence of TDP2. To explain that TOP3Accs remain detectable in the absence of TDP2, may be related to the high levels of TOP3Accs following transfection with R364W-TOP3A, which might saturate the cellular proteolytic activity. Our findings with TOP3Accs are consistent with earlier several studies showing that TDP1 or TDP2 depletion increases TOP1cc, TOP2cc and TOP3Bcc signals (after topoisomerase poison inhibitor treatment or after transfection with self-trapping topoisomerase mutant) by employing ICE/ RADAR assays (Katyal et al., 2014; Chiang et al., 2017; Ghosh et al., 2019, Hoa et al., 2016; Sasanuma et al., 2018). Specifically, TDP1 depletion increases TOP1ccs in *Tdp1*^{-/-} neural tissue (Katyal et al., 2014) and TOP1mtccs in mitochondria (Chiang et al., 2017; Ghosh et al., 2019). Hoa et al., 2016 and Sasanuma et al., 2018 showed that *TDP2*^{-/-} cells accumulate more TOP2ccs by ICE assay compared to *wild-type* cells without the use of proteasome inhibitor (MG132).

Additionally, the TOP3A antibodies might still recognize the cellular TOP3A-DPC polypeptides processed by SPRTN prior to TDP2 excision. This has been clarified experimentally as shown below. We performed ICE assay in HCT116 cells after R364W-TOP3A transfection to examine the proteolytic digestion of TOP3A. After cell lysis, we separated TOP3Accs from free TOP3A in cellular lysates by subjecting them to CsCl-gradient ultracentrifugation-sedimentation. Free TOP3A remained in the top fractions while TOP3Accs moved towards the bottom fractions of CsCl gradient which corresponds to the migration of DNA. TOP3Accs were detected in the middle fractions of the CsCl-gradient as shown in slot blot below. In consistent with RADAR assays results, we confirmed increased TOP3Accs in *TDP2*^{-/-} cells when compared to *wild-type*.

The shifted TOP3Accs signals towards bottom fractions in *TDP2*^{-/-} cells indicate that in the absence of TDP2, degraded TOP3Accs (degraded polypeptides of TOP3A covalently linked to DNA) can be detected using TOP3A antibody. Notably, the TOP3Accs signals seen in the middle fractions were shifted towards the top middle fractions in SPRTN-deficient cells. Thus, the top middle fraction may contain TOP3Accs containing full length TOP3A. Together, SPRTN protease may have partially degraded TOP3Accs, leading to an increase in the specific gravity of this TOP3A-DNA complex and generating signals in the bottom middle fractions.

It is also puzzling that while TDP2 and SPRTN deficiency lead to similar accumulation of TOP3A-R363W cc accumulation, they have a very different effect on the accumulation of wt protein ccs. Are the two proteins recognized and degraded equally?

Response: This point is related to the prior question. At first, SPRTN should recognize and cleave TOP3Accs to degrade most of the protein moiety that is crosslinked to DNA, which we termed ‘Debulking’ and TDP2 then gains access to cleave phosphotyrosyl linkage between DNA and remaining peptides to complete repair process. The accumulation of TOP3Accs in SPRTN-deficient cells even after WT-TOP3A expressing cells suggest that SPRTN might be major and first-acting molecule to degrade TOP3Accs, as proposed in our model (Fig. 6C).

All the experiments with MRE11si might be affected by the replicative problems that its absence may cause. The authors should consider this, and, at least, provide clear evidence that that the effects observed are not due to changes in the replicative status and/or cell cycle stage of the cells.

Response: Thank you for this important concern. Our flow cytometry analysis revealed that siMRE11 U2OS cells has no cell cycle defects in comparison with siControl, which excludes the possibility that the observed effects in MRE11-deficient cells are not due to changes in the replicative status and/or cell cycle stage of the cells. We have included the flow cytometry data of MRE11-deficient cells in Supplementary Fig. 5E of the revised manuscript.

Finally, in order to claim a function of ATM in this process, the authors should include ATM deleted/depleted cells.

Response: As suggested, we corroborated the function of ATM in TOP3Accs repair by silencing ATM. Consistent with the ATM inhibitor data, ATM-depleted cells showed elevated levels of

TOP3Accs in comparison with *wild-type* control cells. These new data have been included in Supplementary Fig. 7B of the revised manuscript.

The last major point is regarding the appropriateness of a model exclusively based on protein over-expression. In Figure 2D, it is clear that TOP3A-overexpressing cells present a highly increased recruitment to the fork. It is therefore important to completely rule out that the effects observed are artefactually derived from the high over-expression conditions, and demonstrate that they truly reflect physiological functions of TOP3A at the fork. At least some of the key experiments should be reproduced with physiological levels of expression of the TOP3A-R364W mutant.

Response: Given that the similar over-expression of WT-TOP3A as R364W-TOP3A had no observed effects on replication and genome stability, we are doubtful that the effects observed with the R367W mutant are artefactually derived from the high over-expression conditions. Furthermore, we also provide additional evidence that siTOP3A cells phenocopy the toxic TOP3A (R364W-TOP3A)-expressing cells regarding replication defects as evidenced by DNA combing assays (Supplementary Figure 3 of the revised manuscript). These results support the requirement of TOP3A activity at replication forks.

As suggested, we tried to generate endogenous bi-allelic TOP3A-R364W mutant in human cells using CRISPR knock-in strategy. We did not obtain any viable colony, consistent with the genotoxicity of overexpressing the R364W mutation.

Additional comments:

- What is the non-nuclear signal in Fig 1D?

Response: The non-nuclear signal in Fig 1D represents mitochondrial signal of TOP3A. This is consistent with prior literature. We confirmed this by immunofluorescence staining of cells with the mitochondrial marker protein, TOMM20 antibody. To clarify this point, we have added a new representative image in Supplementary Figure 1D of the revised manuscript.

- Asymmetric chromatin fibers could also represent broken forks.

Response: Thank you. We cannot exclude this possibility. We excluded chromatin fibers that were probably broken between the bidirectional forks, and we did not even count those as unidirectional fibers.

- How was the clonogenic experiments in Fig 4A quantified? In M&M it indicates that the results are related to NT control, but this is not compatible with the figure.

Response: Thank you. We quantified the clonogenic experiments in Fig. 4A after normalizing to the number of seeded cells at day 0. We have clarified this point and edited the text accordingly in Materials and Methods section of the revised manuscript.

- The increase in Rad51 foci and not only % of colocalization should be shown in Fig 4B-C.

Response: We have added the quantification plot showing the number of Rad51 foci in Fig. 4C of the revised manuscript. Thank you.

REVIEWER COMMENTS

Reviewer #1 (Remarks to the Author):

This manuscript is a milestone in research on TOP3A.

Reviewer #2 (Remarks to the Author):

The authors have done a good job in addressing the majority of my comments and I have no remaining concerns.

As a minor comment, I am still not convinced by the IP experiments showing that SPRTN catalytic activity is required for the interaction with the TOP3A-DPC. To me, it seems more likely that the E112A mutation affects the structure of SPRTN in an indirect way (thereby leading to loss of binding). A better mutant would be E112Q, which was shown to be inactive and is expected to have less severe effects on SPRTN's conformation. It may be good to comment on this issue in the text.

Reviewer #3 (Remarks to the Author):

We appreciate the effort of the authors in addressing our concerns. We consider however that the main issues have not been solved.

As stated in the previous revision, we consider that the main scientific advance of this manuscript is the potential novel function of TOP3A in replication. We do not consider this sufficiently proven, as all the phenotypes observed can be explained by the known, and well characterized, role of TOP3A in dissolving recombination intermediates, which will be formed associated to replication. The authors have produced some novel results in this regard, but we do not consider them to be conclusive. The similarity of phenotypes between TOP3A depletion and expression of TOP3A-R364W is interesting and informative, but does not discriminate between these possibilities, and we completely fail to understand the rationale by which the authors claim that an exposure of ssDNA in nascent strands is indicative of fork reversal. As the authors claim, because of their intimate relationship, it is difficult to separate recombination and replication, but this cannot be a reason to allow an overinterpretation of the results. An informative experiment would be to see how a reduction in homologous recombination (e.g. by Rad51 depletion) affects the replication-dependent accumulation of TOP3Accs.

In addition to this, we still bear specific unresolved concerns on the previous and new experiments presented, as well as some minor points (see below):

Major:

- In all experiments of TOP3A-WT/R364W overexpression, the control should be an empty vector and not mock transfected cells (as we understand that is the case). Otherwise, conclusions on the effect of overexpressing the wild-type protein are difficult to interpret.
- The TOP3A KD experiments are performed with just one siRNA, so off-target effects cannot be ruled out. The same applies to siMRE11, siCTIP and siATM. The SPRTN KD experiments are more solid because of the opposite effect caused by overexpression.
- Fig 4B. The image shown is not representative of the 10% of Rad51-TOP3A-R364W colocalization reported in Fig 4D. With 10% colocalization, is there a real colocalization? Or just coincidence when you have around 10-times more chromatin-associated TOP3A-R364W than wild-type signal?

- The involvement of TRIM41 in TOP3Acc ubiquitylation should be measured directly (DUST), and not with the accumulation of TOP3Accs as an indirect readout.

- Fig 6B and E. These clonogenic survival tests should be related to that of cells overexpressing wild-type TOP3A, otherwise the effects observed may be due to TOP3Acc-independent genetic interactions.

- Fig 7A-C. As mentioned in the first revision, the cell-cycle differential accumulation of TOP3Accs between SPRTN/TDP2 or MRE11 deficiency may be due to a difference in cell cycle progression in that particular experimental setup. The cell cycle stage of the culture at the specific moment that was taken as S and G2 should be included. Otherwise, small differences in the release/progression may confound the results. We appreciate that the authors have included the cell-cycle distribution of siMRE11 cells in asynchronous conditions, but this, although surprising provided that MRE11 is essential for proliferation, is not sufficient to rule out potential artefacts in the synchronization-release experiment. This has therefore not been adequately addressed.

- The experiment to demonstrate that the TOP3A antibody may still recognize proteolyzed fragments (included in the response to referees) is not convincing. The position in the CsCl gradient where the protein-DNA complex migrates is highly variable from sample to sample, and, although it is formally possible, has not been, to the best of my knowledge, proven to correlate with the size of the covalently-bound polypeptide. A direct measurement of this, for example with the DUST assay in the different mutants, would be convincing.

- The question as to why wild-type TOP3Accs accumulate in SPRTN KD (Fig 5B and 6A) but not in TDP2 KO (Fig 6A) has not been solved. No experiment have been performed to address this issue. The fact of SPRTN acting first on the pathway, as the authors suggest, does not explain this, as this would be the same for TOP3A-R364W, for which cleavage-complex accumulation occurs both in SPRTN and TDP2 depletion, displaying an epistatic effect.

- The potential problem of overexpression conditions has not been solved. When a protein is overexpressed, it may act at different places to where it normally does. For example, there may be mechanisms that prevent TOP3A action at replication forks to avoid the problems that it may cause, and these may be overwhelmed when the protein is overexpressed. The authors claim that they have failed to obtain biallelic TOP3A-R364W mutations. However, we understand that the monoallelic mutant could be used to address this. At least to show the accumulation of TOP3Accs and association with the replication fork.

Minor:

- It is unclear why the quantification of the iPOND and chase experiment (Fig 2E) is normalized (in addition it is not indicated as such), and what statistical test has been applied. Please note that a t-test is not appropriate on normalized values, and it is performed like this in other figures.

- For clarity, the use of siControl should be indicated in the figures, and not only in the legends.

- Page 5, line 172. The DNA combing analysis will not detect "replication fork barriers".

- Page 5, line 184. I understand that the authors refer to IdU single-colored fibers.

- Fig 4A. The plates are not shown, just a scheme. The way of calculating the % of colony formation efficiency (from seeded cells) should be indicated in the figure legend.

- Page 9, line 332. A clonogenic assay does not measure repair.

- Page 10, line 347. The genetic relationship between MRE11 and ATM has not been tested.

Reviewer Comments

Reviewer #1 (Remarks to the Author):

This manuscript is a milestone in research on TOP3A.

Response: Thank you very much.

Reviewer #2 (Remarks to the Author):

The authors have done a good job in addressing the majority of my comments and I have no remaining concerns.

Response: Thank you very much.

As a minor comment, I am still not convinced by the IP experiments showing that SPRTN catalytic activity is required for the interaction with the TOP3A-DPC. To me, it seems more likely that the E112A mutation affects the structure of SPRTN in an indirect way (thereby leading to loss of binding). A better mutant would be E112Q, which was shown to be inactive and is expected to have less severe effects on SPRTN's conformation. It may be good to comment on this issue in the text.

Response: Thank you; as suggested, we have added text in the Results section of the revised manuscript: "However, the reduced interaction of SPRTN E112A with TOP3A in chromatin is intriguing because the catalytic residue of proteins is not important for the binding of their substrates. Likely, indirect structural changes of SPRTN due to the E112A mutation might lead to loss of its binding with TOP3A in chromatin."

Reviewer #3 (Remarks to the Author):

We appreciate the effort of the authors in addressing our concerns. We consider however that the main issues have not been solved.

Response: Thank you for your interest in our study and for rigorous analyses and comments, which we have addressed in our second revision.

As stated in the previous revision, we consider that the main scientific advance of this manuscript is the potential novel function of TOP3A in replication. We do not consider this sufficiently proven, as all the phenotypes observed can be explained by the known, and well characterized, role of TOP3A in dissolving recombination intermediates, which will be formed associated to replication. The authors have produced some novel results in this regard, but we do not consider them to be conclusive. The similarity of phenotypes between TOP3A depletion and expression of TOP3A-R364W is interesting and informative, but does not discriminate between these possibilities, and we completely fail to understand the rationale by which the authors claim that an exposure of ssDNA in nascent strands is indicative of fork reversal. As the authors claim, because of their intimate relationship, it is difficult to separate recombination and replication, but this cannot be a reason to allow an overinterpretation of the results. An informative experiment would be to see how a reduction in homologous recombination (e.g., by Rad51 depletion) affects the replication-dependent accumulation of TOP3Accs.

Response: Thank you for finding that a main scientific advance of our study is the novel functions of TOP3A in replication, and that the similarity of phenotypes between expression of TOP3A-R364W and TOP3A depletion is interesting and informative.

Actually, our study elucidates additional novel aspects including the cytotoxic and genotoxicity of persistent TOP3Accs generated by our engineered TOP3A mutant (R364W) that produces replicative damage. We believe that this mutant will enable further studies on TOP3A function not only in the nuclear but also in the mitochondrial genome. Most importantly to us, our study is the first description of the repair pathways removing stalled and potentially toxic TOP3A-DPCs by two different pathways, Spartan-TDP2 and MRE11.

We agree with the reviewer that our prior interpretation of the native BrdU/HU assay was potentially misleading. Indeed, TOP3A-dependent ssDNA in nascent DNA in cells treated with HU, an approach first described by David Cortez (Couch et al., *Genes & Development*, 2013) and recently used by Tian et al. (*Mol. Cell*, 2021) is not a direct evidence of fork reversal as proposed recently by Tian et al. in *Molecular Cell* (2021). Therefore, we have clarified this point in our revised manuscript and stated that BrdU staining in replicating DNA after treatment of cells with HU is indicative of nascent DNA becoming single-stranded possibly as a result of extensively reversed forks.

As suggested, we have investigated the effects of a reduction of the homologous recombination machinery by Rad51 depletion on the accumulation of TOP3Accs. To do so, we depleted RAD51 recombinase in U2OS cells by siRNA transfection and examined the level of TOP3Accs in RAD51-deficient cells after TOP3A-R364W transfection. We found that depleting RAD51 had no impact on the levels of TOP3Accs. We have included these new data in Supplementary Fig. 3 and updated the Results section accordingly. Additionally, it is plausible that TOP3Accs could also be formed while the BTR recognizes and resolves HR intermediates in G2 (Fig. 7D) based on our observations of the detectable level of TOP3Accs in G2 phase (Fig. 2B).

In addition to this, we still bear specific unresolved concerns on the previous and new experiments presented, as well as some minor points (see below):

Major:

- In all experiments of TOP3A-WT/R364W overexpression, the control should be an empty vector and not mock transfected cells (as we understand that is the case). Otherwise, conclusions on the effect of overexpressing the wild-type protein are difficult to interpret.

Response: As suggested, we verified that the effects of empty vector and mock transfection were similar.

- The TOP3A KD experiments are performed with just one siRNA, so off-target effects cannot be ruled out. The same applies to siMRE11, siCTIP and siATM. The SPRTN KD experiments are more solid because of the opposite effect caused by overexpression.

Response: The cocktail of siRNA used against siTOP3A, siMRE11, siCTIP and siATM includes four individual siRNA targeting four different sites in each RNA, rule out their off-target effects. This point has been clarified in the revised manuscript.

- Fig 4B. The image shown is not representative of the 10% of Rad51-TOP3A-R364W colocalization reported in Fig 4D. With 10% colocalization, is there a real colocalization? Or just coincidence when you have around 10-times more chromatin-associated TOP3A-R364W than wild-type signal?

Response: In the figure legend of Fig. 4B, we indicated the percentage of cells displaying TOP3A and RAD51 colocalization. Yes, these are real colocalization. Our interpretation of the relatively low colocalization of RAD51 and TOP3A is that the action of TOP3A to resolve precatenanes is not expected to require RAD51.

- The involvement of TRIM41 in TOP3Acc ubiquitylation should be measured directly (DUST), and not with the accumulation of TOP3Accs as an indirect readout.

Response: Thank you; As suggested, we checked TOP3Acc ubiquitylation level in TRIM41-deficient cells by DUST assay. No reduction in the level of ubiquitylation in TRIM41-deficient cells in comparison with control, indicating that TRIM41 is not E3 ubiquitin ligase for TOP3Accs. We have included these new data in Supplementary Fig. 7E of the revised manuscript.

- Fig 6B and E. These clonogenic survival tests should be related to that of cells overexpressing wild-type TOP3A, otherwise the effects observed may be due to TOP3Acc-independent genetic interactions.

Response: As suggested, in Fig 6B and E, we have represented clonogenic survival histogram of each cell types normalized with their colony forming efficiency in TOP3A-WT transfected condition. This has been clarified in the revised manuscript.

- Fig 7A-C. As mentioned in the first revision, the cell-cycle differential accumulation of TOP3Accs between SPRTN/TDP2 or MRE11 deficiency may be due to a difference in cell cycle progression in that particular experimental setup. The cell cycle stage of the culture at the specific moment that was taken as S and G2 should be included. Otherwise, small differences in the release/progression may confound the results. We appreciate that the authors have included the cell-cycle distribution of siMRE11 cells in asynchronous conditions, but this, although surprising provided that MRE11 is essential for proliferation, is not sufficient to rule out potential artefacts in the synchronization-release experiment. This has therefore not been adequately addressed.

Response: The residual amount of MRE11 after siMRE11 transfection might be enough to support cellular proliferation. Moreover, the absence of cell cycle defects in siMRE11 U2OS cells is consistent with a recent study showing that shMRE11 cells had normal cell cycle distributions in human VM-CUB1 urothelial cells (Na et al., Cell Death Dis, 2021). Additionally, lack of expression of MRE11 is well-established in colon tumors with mismatch repair deficiency (our prior work: Takemura et al. 2006 JBC).

We did provide flow cytometry data in our previously revised manuscript showing that the synchronization is comparable between cells expressing TOP3A WT vs R364W. The data are included in Supplementary Fig. 2D. Regarding the cell cycle dependency of TOP3Acc repair, we cannot totally exclude that the effect of SPRTN, MRE11 and TDP2 of cell cycle progression may contribute to additional indirect repair defects. This point has been mentioned at the end of the Discussion. Thank you.

- The experiment to demonstrate that the TOP3A antibody may still recognize proteolyzed fragments (included in the response to referees) is not convincing. The position in the CsCl gradient where the protein-DNA complex migrates is highly variable from sample to sample, and, although it is formally possible, has not been, to the best of my knowledge, proven to correlate with the size of the covalently-bound polypeptide. A direct measurement of this, for example with the DUST assay in the different mutants, would be convincing.

Response: We believe that ICE assay using CsCl-gradient ultracentrifugation can detect protein-DNA complexes with differential polypeptide lengths covalently linked to DNA, which is not the case for the DUST assay. The position of the protein-DNA complexes in CsCl gradient reflects their size as shown by Sasanuma et al, PNAS, 2018 for TOP2ccs.

- The question as to why wild-type TOP3Accs accumulate in SPRTN KD (Fig 5B and 6A) but not in TDP2 KO (Fig 6A) has not been solved. No experiment have been performed to address this issue. The fact of SPRTN acting first on the pathway, as the authors suggest, does not explain this, as this would be the same for TOP3A-R364W, for which cleavage-complex accumulation occurs both in SPRTN and TDP2 depletion, displaying an epistatic effect.

Response: The accumulation of TOP3Accs after wild-type TOP3A transfection in SPRTN knock-down but not in TDP2 KO cells is consistent with the previously published findings that SPRTN knock-down cells display accumulation of endogenous TOP1ccs and TOP2ccs (Vaz B et al., Mol Cell, 2016). Our findings imply that SPRTN is a rate limiting step for TOP3Acc processing. Yet, after TOP3A-R364W transfection, the observed increase of TOP3Accs upon TDP2 depletion confirms that the proteolysis by SPRTN is not sufficient to remove TOP3Accs in the absence of TDP2. The fact that TOP3Accs remain detectable in the absence of TDP2, may be related to the high levels of TOP3Accs following transfection with R364W-TOP3A, which might saturate the cellular proteolytic activity. Our findings with TOP3Accs are consistent with earlier studies showing that TDP1 or TDP2 depletion increases TOP1cc, TOP2cc and TOP3Bcc signals (after topoisomerase poison inhibitor treatment or after transfection with self-trapping topoisomerase mutant) by employing ICE/ RADAR assays (Katyal et al., 2014; Chiang et al., 2017; Ghosh et al., 2019, Hoa et al., 2016; Sasanuma et al., 2018)

- The potential problem of overexpression conditions has not been solved. When a protein is overexpressed, it may act at different places to where it normally does. For example, there may be mechanisms that prevent TOP3A action at replication forks to avoid the problems that it may cause, and these may be overwhelmed when the protein is overexpressed. The authors claim that they have failed to obtain biallelic TOP3A-R364W mutations. However, we understand that the monoallelic mutant could be used to address this. At least to show the accumulation of TOP3Accs and association with the replication fork.

Response: WT-TOP3A and R364W-TOP3A were similarly expressed after transfection in all cells examined in this study (Supplementary Fig. 1), and we did not observe replication defects and genome instability in TOP3A-WT-overexpressing cells. Thus, we are confident that the results observed with the R364W mutant are not due to high over-expression conditions.

This point is indicated in the manuscript

Minor:

- It is unclear why the quantification of the iPOND and chase experiment (Fig 2E) is normalized (in addition it is not indicated as such), and what statistical test has been applied. Please note that a t-test is not appropriate on normalized values, and it is performed like this in other figures.

Response: Thank you. We performed the quantification of chase signal as suggested by Reviewer 1 during previous revision. We normalized the chase signal with click signal of TOP3A as well as H3 signal as described in the figure legends of Fig. 2E. As indicated in figure legends, P values were obtained by applying paired t-test. ** = $p < 0.01$, which is commonly used in published studies.

- For clarity, the use of siControl should be indicated in the figures, and not only in the legends.

Response: Thank you. As suggested, we have indicated siControl in the figures (Figs. 5 and 7).

- Page 5, line 172. The DNA combing analysis will not detect “replication fork barriers”.

Response: We agree and have corrected the sentence as “To assess the stalling or collapse of replication fork in TOP3A-R364W-expressing cells” in place of “To assess the presence of replication fork barriers in TOP3A-R364W-expressing cells”.

- Page 5, line 184. I understand that the authors refer to IdU single-colored fibers.

Response: Thank you. We have replaced “CldU” with “IdU” in the revised manuscript.

- Fig 4A. The plates are not shown, just a scheme. The way of calculating the % of colony formation efficiency (from seeded cells) should be indicated in the figure legend.

Response: Sorry, the images were included in the previously submitted Figure 4 file. Unfortunately, they were not visible in the merged manuscript file after the Journal’s automated file conversion.

The sentence “Percent of colony formation efficiency was calculated as the percentage of surviving cells after TOP3A plasmids transfection relative to the number of seeded cells of each cell types” has been added in the legend of Fig. 4A of the revised manuscript.

- Page 9, line 332. A clonogenic assay does not measure repair.

Response: Indeed. Thank you for pointing this oversight. We have rephrased the sentence and replaced “to repair TOP3Accs-induced DNA damage” with “to act on TOP3Accs-induced DNA damage response” in the revised manuscript.

- Page 10, line 347. The genetic relationship between MRE11 and ATM has not been tested.

Response: As suggested, we have tested the genetic relationship between MRE11 and ATM in TOP3Accs repair. We found that the ATM inhibition in MRE11-depleted cells caused no further increase in TOP3Accs level in comparison with ATM inhibited and MRE11-deficient cells alone, suggesting an epistasis relationship of ATM and MRE11. We have included these data in Supplementary Fig. 8C of the revised manuscript.

REVIEWERS' COMMENTS

Reviewer #3 (Remarks to the Author):

The authors have addressed my main concern, showing that the TOP3Accs are HR-independent. So, conceptually, I have no further comment.

There are still, however, technical issues that, in my opinion, cannot be overlooked.

- Using a cocktail of 4 siRNAs does not rule out off-target effect, but quite the opposite (increase it 4-fold).

- No evidence is provided that the siRNA-depleted cells in Fig7A-C are in the cell cycle stage indicated. This control is required. You cannot indicate that the siRNA-transfected cells are in a specific cell cycle stage just by presuming that they will be released and progress with the same kinetics than wild-type cells (shown in Supplementary Fig. 2D). And it is a simple experiment to perform.

- A t-test (paired or not) is not appropriate for normalized values. I am reviewing this manuscript and not other published studies. For clarification please see Nat Methods 2011;8:104-5

REVIEWERS' COMMENTS

Reviewer #3 (Remarks to the Author):

The authors have addressed my main concern, showing that the TOP3Accs are HR-independent. So, conceptually, I have no further comment.

Response: Thank you very much.

There are still, however, technical issues that, in my opinion, cannot be overlooked.

- Using a cocktail of 4 siRNAs does not rule out off-target effect, but quite the opposite (increase it 4-fold).

Response: As per reviewer's concern about off-target effect of siRNA, as a representative, we have tested the effect of knocking down of MRE11 and TDP2 on the level of TOP3Accs using additional siRNA (source: OriGene) targeting MRE11 and TDP2, respectively. We have found an increased level of TOP3Acc in MRE11- and TDP2-depleted cells in comparison with siControl cells, which is consistent with our previous observations using SMARTPool (cocktail of 4 siRNAs;source: Horizon Discovery, Dharmacon). We have added this data here as follows, not in the manuscript.

These observations rule out the possibility of the off-target effect of siRNA that we used previously.

- No evidence is provided that the siRNA-depleted cells in Fig7A-C are in the cell cycle stage indicated. This control is required. You cannot indicate that the siRNA-transfected cells are in a specific cell cycle stage just by presuming that they will be released and progress with the same kinetics than wild-type cells (shown in Supplementary Fig. 2D). And it is a simple experiment to perform.

Response: We have provided cell cycle synchronization data related to Fig.7A-C to ensure all cell types examined were in the same stages of cell cycle. This data has been included in the Supplementary Fig. 9A of the revised manuscript.

- A t-test (paired or not) is not appropriate for normalized values. I am reviewing this manuscript and not other published studies. For clarification, please see Nat Methods 2011;8:104-5

Response: Thank you for the clarification provided with the reference article above. As suggested, we have performed new statistical analyses and included the exact p-value of these analyses in the figure legends of the revised manuscript.

This referenced article stated that “Such normalization can be performed in two ways (Fig. 1a). In the first approach, all values of control and treatment samples are divided by the mean of the control sample, which thus becomes the arbitrary reference value ('normalization i'). Such normalization conserves the distribution and the relative variance of the samples, allowing the subsequent use of a *t*-test. A second way to perform the normalization is to divide the control and treatment values from each experimental run by the control value ('normalization ii'). Such normalization converts the distribution of the control sample into a uniform distribution with zero variance and renders the normalized data unsuitable for a *t*-test.”

In the revised manuscript, we followed the first approach using 'normalization i'. Following the above article, as reviewer suggested, we have performed statistical analysis using ANOVA for multiple (more than two) grouped analyses, not t-test. For normalized values where two groups used, we followed the first approach using 'normalization i' and performed t-test with Welch's correction such as for Fig. 2f. As expected, the p-values of these figures were changed, however, the statistical significance remained the similar as compared to our previous analyses.